# VJEPA: Variational Joint Embedding Predictive Architectures as Probabilistic World Models

**Yongchao Huang** [1]

## Abstract

Joint Embedding Predictive Architectures (JEPAs) avoid pixel reconstruction by predicting latent representations, but standard formulations remain deterministic and provide limited uncertainty estimates for planning and control. We introduce *Variational JEPA (VJEPA)*, a probabilistic extension that learns predictive distributions over future latent states using a latent-space variational objective, without autoregressive observation likelihoods. We show that VJEPA links JEPA-style self-supervised learning to predictive state representations and Bayesian filtering, and that its latent variables can serve as sufficient information states for control when they preserve task-relevant predictive information. We also propose *Bayesian JEPA (BJEPA)*, which combines a learned dynamics expert with modular prior experts through a Product of Experts, enabling constraint-aware prediction and zero-shot prior swapping. Tests on Noisy-TV systems, nonlinear and image-based benchmarks, STL-10 with a ViT encoder, and DMC Cheetah-run show that predictive JEPA-family objectives are more robust to high-variance nuisance distractors than reconstruction-based world-model baselines, which position VJEPA as a principled framework for robust, uncertainty-aware, reconstruction-free world models.

## 1. Introduction

Self-supervised representation learning has largely been shaped by two paradigms: generative modeling and contrastive learning. Generative approaches optimize likelihoods or reconstruction objectives over high-dimensional observations (Kingma & Welling, 2022; van den Oord et al., 2016), which can force representations to encode task-irrelevant, high-entropy details such as background noise or stochastic distractors. Contrastive methods (Chen et al., 2020; He et al., 2020) avoid reconstruction, but often rely on negative sampling, augmentation design, or instance-discrimination assumptions that may introduce their own biases. Joint Embedding Predictive Architectures (JEPAs) (Assran et al., 2023) provide a third route: learning by predicting *representations* of missing or future data rather than reconstructing observations. By operating in latent space, JEPAs avoid pixel-level likelihoods and can focus capacity on semantically or dynamically predictive structure. Recent instantiations, including I-JEPA (Assran et al., 2023), V-JEPA (Bardes et al., 2024) and C-JEPA (Mo & Tong, 2024) etc, have demonstrated the effectiveness of this principle in image and video representation learning.

However, most existing JEPA formulations remain largely *deterministic*. They train predictors through regression losses between latent embeddings, which provides limited probabilistic semantics and does not directly represent uncertainty over future latent states. This limitation becomes important when using JEPA-style models as world models for planning and control: stochastic environments may admit multiple plausible futures, and decision-making requires knowing not only what is predicted, but also how uncertain the prediction is. Moreover, the conditions under which a learned latent embedding can serve as a sufficient information state for control remain underexplored.

In this work, we formulate JEPA as a probabilistic predictive state-space model. Our contributions are as follows.

(1) **Variational JEPA (VJEPA).** We introduce a variational extension of JEPA that learns a predictive distribution over future latent states. The resulting objective retains the reconstruction-free nature of JEPA while enabling uncertainty estimation in latent space, without requiring autoregressive observation likelihoods.

(2) **Information-theoretic analysis.** We relate the VJEPA objective to a variational lower bound on predictive mutual information. This analysis explains why latent predictive objectives can preserve information relevant to future prediction while discarding high-variance nuisance variation

[1]Department of Computing Science, University of Aberdeen, Aberdeen, UK. Correspondence to: Yongchao Huang <yongchao.huang@abdn.ac.uk>.

*Proceedings of the 43rd International Conference on Machine Learning*, Seoul, South Korea. PMLR 306, 2026. Copyright 2026 by the author(s).

that reconstruction-based objectives are forced to model.

(3) **Predictive sufficiency for control.** We show that, under predictive sufficiency assumptions, VJEPA representations can serve as sufficient information states for optimal control. This establishes that a valid latent world model need not reconstruct pixels, provided its latent state preserves the predictive information needed for future decision-making.

(4) **Bayesian JEPA (BJEPA).** We propose a modular Bayesian extension that factorizes predictive belief into a learned dynamics expert and a structural prior expert. Combining these experts through a Product of Experts enables constraint-aware prediction, zero-shot prior swapping, and task adaptation without retraining the dynamics model.

Empirically, we evaluate these ideas on controlled Noisy-TV systems, nonlinear and image-based benchmarks, natural-image tests with a ViT encoder, and a continuous-control model-based RL task. Across these, JEPA-family predictive objectives are more robust to high-variance nuisance distractors than reconstruction-based baselines, supporting probabilistic latent prediction as a principled route to robust, uncertainty-aware, reconstruction-free world models.

## 2. Related Work

**Latent dynamics and world models.** Model-based reinforcement learning has extensively used latent dynamics for planning and policy learning. Methods such as PlaNet (Hafner et al., 2019) and Dreamer (Hafner et al., 2020) learn probabilistic transitions $p(z_{t+1} \mid z_t, a_t)$ together with observation models $p(x_t \mid z_t)$, typically by optimizing an evidence lower bound (ELBO) involving reconstruction or likelihood terms in observation space. While highly effective, such reconstruction-based objectives can encourage the latent state to encode task-irrelevant, high-entropy variability, such as visual distractors or background noise. MuZero (Schrittwieser et al., 2020) avoids explicit observation reconstruction, but learns its latent dynamics through task-specific reward, value, and policy supervision. VJEPA differs from both lines of work by learning probabilistic latent dynamics through self-supervised representation prediction, without requiring pixel reconstruction, reward labels, or policy optimization during representation learning.

**Joint Embedding Predictive Architectures.** Joint Embedding Predictive Architectures (JEPAs) (Assran et al., 2023) shift the self-supervised learning objective from reconstructing observations to predicting latent representations of missing or future data. Instantiations such as I-JEPA (Assran et al., 2023) and V-JEPA (Bardes et al., 2024) show that prediction in embedding space can yield strong semantic and temporal representations while avoiding pixel-level likelihoods. This idea is closely related to LeCun's broader vision of Hierarchical JEPA (H-JEPA) as a foundation for world

modeling (LeCun, 2022). Recent JEPA-based world-model and robotic-control studies (Terver et al., 2026; Assran et al., 2025) further suggest that latent-space prediction can support planning. However, most existing JEPA formulations rely on deterministic regression objectives. As a result, the probabilistic interpretation of JEPA, the representation of predictive uncertainty, and the conditions under which JEPA embeddings form sufficient information states for control remain underdeveloped.

**Predictive state representations and active inference.** VJEPA connects representation learning to classical ideas in control and probabilistic filtering. Predictive State Representations (PSRs) define state in terms of predictions about future observations or tests rather than latent physical variables (Littman & Sutton, 2001; Singh et al., 2004). In this sense, VJEPA can be viewed as a neural, amortized PSR in which the representation summarizes the information needed to predict future latent states. VJEPA is also conceptually related to Active Inference (Friston, 2010), where perception and belief updating are framed as variational inference in a generative model. The key difference is that VJEPA performs predictive belief propagation entirely in representation space: it does not require a sensory likelihood $p(x \mid z)$ or pixel-level reconstruction objective. This yields a reconstruction-free route to probabilistic latent world modeling while retaining connections to filtering, predictive sufficiency, and uncertainty-aware planning.

## 3. VJEPA: Variational Formulation of JEPA

**Background of JEPA.** Given a context-target pair $(x_C, x_T)$, a context encoder $f_\theta$ computes $Z_C = f_\theta(x_C)$ and a target encoder $f_{\bar\theta}$ (typically an EMA copy of $f_\theta$, updated as $\bar\theta \leftarrow \tau\bar\theta + (1-\tau)\theta$) provides $Z_T = f_{\bar\theta}(x_T)$; a predictor $g_\phi$ outputs $\hat{Z}_T = g_\phi(Z_C, \xi_T)$ where $\xi_T$ is structural side information. Standard JEPA minimizes

$$\mathcal{L}_{\text{JEPA}} = \|\hat{Z}_T - Z_T\|^2 \tag{1}$$

which is deterministic and cannot express uncertainty over future latent states. We can give this standard deterministic JEPA a probabilistic interpretation. The common squared-loss objective admits a simple probabilistic interpretation: it is equivalent, up to an additive constant and a positive scale factor, to maximizing the likelihood of $Z_T$ under a fixed-variance isotropic Gaussian predictive model,

$$-\log \mathcal{N}(Z_T \mid \hat{Z}_T, \sigma^2 I) = \frac{1}{2\sigma^2}\|\hat{Z}_T - Z_T\|^2 + \frac{d}{2}\log(2\pi\sigma^2). \tag{2}$$

Thus, deterministic JEPA can be viewed as using an implicit fixed-uncertainty latent predictive model. This interpretation is useful but limited in stochastic environments, where multiple future latent states may be plausible and where uncertainty itself is important for filtering, planning, control.

VJEPA generalizes this fixed-variance view by replacing point prediction with an explicit predictive distribution

$$p_\phi(Z_T \mid Z_C, \xi_T), \tag{3}$$

where $Z_C = f_\theta(x_C)$ is the context representation and $\xi_T$ denotes structural side information such as mask position, temporal offset, or action information.

### 3.1. Probabilistic Predictive Model

Given a context-target pair $(x_C, x_T)$, the context branch computes

$$Z_C = f_\theta(x_C), \tag{4}$$

while the target branch defines a stochastic target representation

$$q_{\bar\theta}(Z_T \mid x_T). \tag{5}$$

Here $\bar\theta$ denotes target-branch parameters, typically updated as an EMA of the online encoder parameters. The predictor is a neural network that outputs the parameters of a conditional latent distribution,

$$p_\phi(Z_T \mid Z_C, \xi_T), \tag{6}$$

for example a diagonal Gaussian with learned mean and variance:

$$p_\phi(Z_T \mid Z_C, \xi_T) = \mathcal{N}\left(Z_T; \mu_\phi(Z_C, \xi_T), \mathrm{diag}(\sigma_\phi^2(Z_C, \xi_T))\right). \tag{7}$$

This makes VJEPA a probabilistic latent predictor: uncertainty is represented by predicted latent distribution, while the model remains likelihood-free w.r.t. raw observations.

**Deterministic JEPA as a limiting case.** If the target distribution is treated as a point mass at $f_{\bar\theta}(x_T)$ and the predictor is a fixed-variance Gaussian centered at a deterministic predictor $g_\phi(Z_C, \xi_T)$, then the predictive negative log-likelihood reduces to the standard squared JEPA loss up to constants. In this sense, deterministic JEPA is recovered as a fixed-uncertainty limiting case of latent probabilistic prediction. A feature-by-feature comparison between deterministic JEPA and VJEPA is given in Table 4 (Appendix.A).

### 3.2. The Variational Objective

VJEPA is trained entirely in latent space. Given a context-target pair $(x_C, x_T)$ and side information $\xi_T$, the target encoder defines a distribution $q_{\bar\theta}(Z_T \mid x_T)$ and the predictor defines $p_\phi(Z_T \mid Z_C, \xi_T)$ with $Z_C = f_\theta(x_C)$. The VJEPA objective is

$$\mathcal{L}_{\mathrm{VJEPA}} =$$
$$\mathbb{E}_{(x_C, x_T, \xi_T) \sim p_{\mathrm{data}}} \mathbb{E}_{Z_T \sim q_{\bar\theta}(\cdot \mid x_T)} \left[ -\log p_\phi(Z_T \mid f_\theta(x_C), \xi_T) \right]$$
$$+ \beta \, \mathbb{E}_{x_T \sim p_{\mathrm{data}}} \mathrm{KL}\left( q_{\bar\theta}(Z_T \mid x_T) \,\|\, p_0(Z_T) \right), \tag{8}$$

where $p_0$ is a fixed latent prior, typically $\mathcal{N}(0, I)$.

The first term is a latent negative log-likelihood. It trains the context encoder and probabilistic predictor to match the target-side latent distribution without reconstructing observations. The second term is a target-side KL regularizer: it anchors the target latent distribution to a simple prior and helps stabilize the target representation space. This KL should be interpreted as regularization of the target latent space, rather than as an explicit information bottleneck on the context representation.

In implementations where the target branch is a strict EMA reference, the KL regularizer can be evaluated on the corresponding online target-side distribution before the EMA update; for notational simplicity, we write the stabilized target-side distribution as $q_{\bar\theta}$. The essential point is that the KL term regularizes the target latent family, whereas the predictive NLL trains the context-to-target latent predictor.

The predictive term also has an information-theoretic interpretation. Under the induced population distribution over $(Z_C, \xi_T, Z_T)$, the latent NLL is a conditional cross-entropy between the induced target conditional and the variational predictor $p_\phi$. Therefore, minimizing it can be interpreted as maximizing a Barber–Agakov-style lower bound on the predictive mutual information between context and target latents, provided the target-latent marginal entropy is controlled. This motivates the use of VJEPA as a predictive latent-state model for uncertainty-aware planning.

The full architecture is visualized in Figure 1.

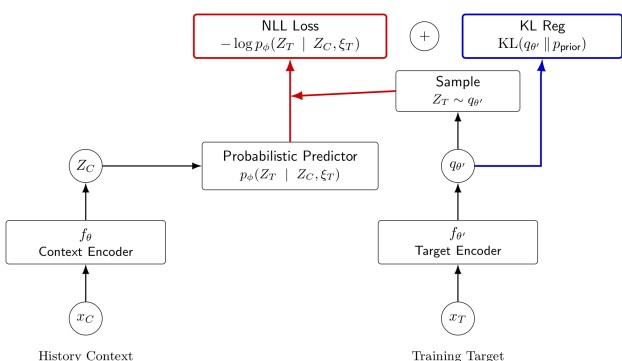

*Figure 1.* VJEPA architecture. Context $x_C$ is encoded into $Z_C$ and used to predict a distribution over the target latent $Z_T$. The target encoder provides a stochastic target distribution $q_{\bar\theta}(Z_T \mid x_T)$, while the target-side KL regularizer stabilizes latent target space.

*Remark* 3.1 (Gaussian implementation). Both $q_{\bar\theta}$ and $p_\phi$ are parameterized as diagonal Gaussians, giving closed-form KL and NLL terms. More expressive families (GMMs, diffusion heads) each require a modified variational or likelihood objective and are not drop-in substitutions; a preliminary $K{=}2$ GMM experiment is in Appendix.I.

### 3.3. Training Algorithm

The training loop encodes context, computes the target-side latent distribution, evaluates the predictive NLL and target-side KL, updates encoder and predictor by gradient descent, and updates the target branch by EMA. Full pseudocode is given in Algorithm 1 (Appendix.B).

### 3.4. Collapse Avoidance

Unlike contrastive methods, VJEPA does not require negative samples. Its predictive objective discourages constant-output collapse whenever the context contains information useful for predicting the target latent. The following result formalizes this idealized global-optimum intuition.

**Theorem 3.2** (Constant-context collapse is not globally optimal). *Assume that the target branch is non-collapsed and that the context contains predictive information about the target latent:*

$$I(Z_T; X_C \mid \xi_T) > 0, \qquad Z_T \sim q_{\bar\theta}(\cdot \mid X_T). \quad (9)$$

*Assume also that the predictor class is sufficiently expressive to represent the relevant conditional distributions. Then a representation satisfying $f_\theta(X_C) \equiv c$ cannot be a global minimizer of the predictive NLL term in Equation (8).*

*Proof sketch.* If $f_\theta(X_C) \equiv c$, then the predictor cannot use the context and the best achievable predictive NLL is the conditional entropy $H(Z_T \mid \xi_T)$. If the context representation preserves the predictive information in $X_C$, the best achievable predictive NLL is $H(Z_T \mid X_C, \xi_T)$. The gap is

$$H(Z_T \mid \xi_T) - H(Z_T \mid X_C, \xi_T) = I(Z_T; X_C \mid \xi_T), \quad (10)$$

which is strictly positive by assumption. Therefore, a constant context representation incurs an irreducible predictive penalty and cannot be globally optimal. □

*Remark* 3.3 (Scope of the guarantee). Theorem 3.2 rules out only constant-output collapse of the context representation under idealized assumptions. It does not guarantee that finite-sample optimization avoids all degenerate solutions, nor does it by itself prevent collapse of the target branch. In practice, EMA targets, architectural asymmetry, target-side KL regularization, variance/covariance regularization, or VICReg-style penalties may still be useful for stable training. □

## 4. Information-Theoretic Analysis

We analyze VJEPA through lens of information theory, relating the variational objective to maximization of predictive mutual information and Predictive Information Bottleneck (PIB) principle (Bialek et al., 2001; Tishby et al., 2000).

### 4.1. Variational Maximization of Predictive Information

We establish that minimizing the VJEPA loss maximizes a lower bound on the mutual information between $Z_t$ and $Z_{t+\Delta}$.

**Theorem 4.1** (Variational MI Lower Bound). *Let $(Z_t, Z_{t+\Delta})$ be the joint distribution of context and target representations. The mutual information $I(Z_t; Z_{t+\Delta})$ is lower-bounded by the negative cross-entropy (expected log-likelihood) of the predictive distribution:*

$$I(Z_t; Z_{t+\Delta}) \geq \mathbb{E}\big[\log p_\phi(Z_{t+\Delta} \mid Z_t)\big] + H(Z_{t+\Delta}). \quad (11)$$

*Proof.* By definition, $I(Z_t; Z_{t+\Delta}) = H(Z_{t+\Delta}) - H(Z_{t+\Delta} \mid Z_t)$. The conditional entropy is $H(Z_{t+\Delta} \mid Z_t) = -\mathbb{E}[\log p(Z_{t+\Delta} \mid Z_t)]$. Using the non-negativity of KL divergence $D_{\mathrm{KL}}(p(\cdot|Z_t) \,\|\, p_\phi(\cdot|Z_t)) \geq 0$, we have $\mathbb{E}[\log p(Z_{t+\Delta} \mid Z_t)] \geq \mathbb{E}[\log p_\phi(Z_{t+\Delta} \mid Z_t)]$. Substituting this yields the result (Barber-Agakov bound (Barber & Agakov, 2003)). □

**Implication.** The VJEPA objective (Eq. 8) minimizes $-\log p_\phi$. Since $H(Z_{t+\Delta})$ depends only on the target encoder (which evolves slowly), VJEPA effectively maximizes the *mutual information* between the past and future representations.

### 4.2. Nuisance Invariance

The Predictive Information Bottleneck (PIB) principle (Bialek et al., 2001) seeks a representation of the past $Z_t$ that is maximally predictive of the future $Z_{t+\Delta}$ while discarding irrelevant details.

**Proposition 4.2** (Invariance to Nuisance Variability). *Let input $x = S + N$, where $S$ is signal and $N$ is nuisance noise independent of the future target. Generative models maximizing $p(x|Z)$ must encode $N$ to minimize reconstruction error. In contrast, the VJEPA objective is invariant to representations that discard $N$, provided they preserve $I(Z_t; Z_{t+\Delta})$.*

(Proof in Appendix.F). This implies that VJEPA *allows* the representation to be minimal (compressing away noise), whereas reconstruction *forces* the representation to be maximal (retaining noise).

*Remark* 4.3. Proposition 4.2 establishes a formal contrast between predictive and *generative/reconstruction* objectives; it is **not** a claim of strict finite-sample dominance over deterministic JEPA. The proposition states that VJEPA *permits* the encoder to discard nuisance noise without incurring a training penalty; whether a particular run exploits this freedom depends on model capacity, optimisation trajectory, and the signal-to-noise ratio. A well-tuned deterministic JEPA can match or exceed VJEPA at moderate distractor

levels (see Table 3 results); the probabilistic formulation's advantage is most pronounced in high-variance regimes and when calibrated uncertainty estimates are required (e.g., for risk-sensitive planning or anomaly detection).

### 4.3. Contrast with Generative Objectives

We can formally distinguish VJEPA from autoregressive (AR) world models by decomposing their objectives. Generative models minimize $\mathcal{L}_{\text{AR}} \approx -I(Z_t; S_{t+\Delta}) - I(Z_t; N_{t+\Delta})$. The second term is a penalty: the loss increases if the model fails to predict the noise. VJEPA minimizes $\mathcal{L}_{\text{VJEPA}} \approx -I(Z_t; S_{t+\Delta})$. Because the target $Z_{t+\Delta}$ is an abstract representation that can filter noise, there is no penalty for ignoring $N_{t+\Delta}$. This allows the model to focus capacity on the causal dynamics relevant for planning.

## 5. JEPA as a Latent Dynamical System

While standard JEPA predicts spatial patches (I-JEPA) or clip-level features (V-JEPA), we specialize VJEPA to *time-indexed* sequences to form a world model. This distinction is crucial: V-JEPA learns strong temporal features but does not necessarily enforce a compositional state transition model that can be iterated. Our formulation defines a rigorous latent dynamical system suitable for control.

*Table 1.* Comparison of JEPA variants.

| Variant | Prediction | Structure | Dynamical? |
|---|---|---|---|
| I-JEPA | Masked regions | Spatial/semantic | ✗ |
| V-JEPA | Masked clips | Spatiotemporal | ✗ |
| **Time-indexed** | $Z_{t+\Delta}$ from $Z_t$ | State Transition | ✓ |

### 5.1. Latent Transition Model

Let the context be history $x_{\leq t}$ and the target be future $x_{t+\Delta}$. The encoder defines a latent state $Z_t = f_\theta(x_{\leq t})$. The probabilistic predictor then acts as a *latent transition kernel*:

$$Z_{t+\Delta} \sim p_\phi(Z_{t+\Delta} \mid Z_t, u_{t:t+\Delta-1}, \xi_{t+\Delta}). \quad (12)$$

This formulation yields a probabilistic state-space model that is *sequential* but not *autoregressive* in observation space. Unlike generative models which factorize $p(x_{1:T}) = \prod p(x_t \mid x_{<t})$, VJEPA supports belief propagation via Monte Carlo sampling of latent trajectories without ever estimating pixel densities:

$$p(Z_{t+1} \mid u_t) = \int p_\phi(Z_{t+1} \mid Z_t, u_t) \, p(Z_t) \, dZ_t. \quad (13)$$

This allows the model to represent multi-modal futures (e.g., a robot facing a fork in the road) purely in embedding space.

### 5.2. Connection to Classical Bayesian Filters

The latent transition model defined above is not merely a notational convenience; it precisely recovers two classical filtering architectures as special cases, providing a rigorous grounding for VJEPA's uncertainty estimates.

**Kalman-filter analogy.** In the linear-Gaussian regime ($f_\theta$ linear, $p_\phi$ Gaussian with linear mean), the VJEPA predictor $p_\phi(Z_{t+1} \mid Z_t, u_t) = \mathcal{N}(\mu_\phi(Z_t, u_t), \Sigma_\phi(Z_t, u_t))$ generalises the classical Kalman prediction step

$$\mu_{t+1|t} = A\mu_t + Bu_t, \qquad P_{t+1|t} = AP_t A^\top + Q, \quad (14)$$

in two directions: (i) $\mu_\phi$ replaces the fixed matrices $A$, $B$ with a learned nonlinear transition; and (ii) $\Sigma_\phi$ replaces the constant process noise $Q$ with a *heteroscedastic*, state-dependent covariance, so the model can express high uncertainty in chaotic regions and low uncertainty in stable ones. The Kalman *correction* step (measurement update via Kalman gain) is **amortised**: rather than an explicit gain computation, the context encoder $f_\theta$ maps the updated history $x_{\leq t+1}$ directly to the posterior state $Z_{t+1}$, learning implicit Bayesian updates from data. The VJEPA training objective, which minimizes the divergence between the predicted belief $p_\phi(\cdot \mid Z_t)$ and the target-encoder distribution $q_{\theta'}(\cdot \mid x_{t+1})$-drives the predictor to be consistent with these implicit updates. Full derivation and an empirical comparison against an oracle KF in Appendix.H.

**Particle-filter interpretation.** For non-Gaussian or multi-modal beliefs, VJEPA supports sequential importance resampling. Let the belief at time $t$ be a set of $K$ weighted particles $\{Z_t^{(k)}, w_t^{(k)}\}_{k=1}^K$. *Prediction:* propagate each particle via $Z_{t+1}^{(k)} \sim p_\phi(Z_{t+1} \mid Z_t^{(k)}, u_t)$. *Update:* when observation $x_{t+1}$ arrives, the target encoder $q_{\theta'}(Z_{t+1} \mid x_{t+1})$ acts as a surrogate measurement model; the unnormalized weight becomes

$$\tilde{w}_{t+1}^{(k)} \propto w_t^{(k)} \frac{q_{\theta'}(Z_{t+1}^{(k)} \mid x_{t+1})}{p_{\text{ref}}(Z_{t+1}^{(k)})},$$

where $p_{\text{ref}}$ is the KL reference prior from the VJEPA objective. After normalisation and resampling, the particles approximate $p(Z_{t+1} \mid x_{1:t+1})$ *entirely in representation space* and without pixel-level likelihoods. Both filtering strategies are compatible with the VJEPA-MPC loop in Appendix.G.3; see Appendix.H for the full algorithm.

### 5.3. Predictive Sufficiency for Control

In Partially Observable Markov Decision Processes (POMDPs), a Bayesian belief state is sufficient for optimal control (Åström, 1965; Kaelbling et al., 1998). VJEPA does not compute a posterior over environment states $s_t$; instead, it learns a *predictive information state* directly.

**Definition 5.1** (Predictive Sufficiency). A representation $Z_t$ is predictively sufficient if the conditional distribution of future latent trajectories (and associated costs) depends on history $h_t$ only through $Z_t$:

$$p(Z_{t+1:H} \mid h_t, u_{t:H}) = p(Z_{t+1:H} \mid Z_t, u_{t:H}).$$

**Theorem 5.2** (Optimal Control Sufficiency). *If $Z_t$ is predictively sufficient for the planning horizon, and the task cost $c_t$ is measurable w.r.t $Z_t$, there exists an optimal policy $\pi^\star(u_t \mid h_t)$ that depends on history only through $Z_t$: $\pi^\star(u_t \mid Z_t)$.*

*Proof sketch.* See Appendix.G. If $Z_t$ satisfies the definition, it serves as a sufficient statistic for the future. The control problem formally reduces to a fully observable MDP defined over the latent space $\mathcal{Z}$. Standard dynamic programming arguments apply. □

### 5.4. Latent-Space Planning (VJEPA-MPC)

Leveraging this sufficiency, we perform Model Predictive Control (MPC) entirely in the latent space. The objective is to minimize the expected cumulative cost over horizon $H$:

$$u^\star_{0:H-1} = \arg\min_u \mathbb{E}_{Z \sim p_\phi} \left[ \sum_{k=0}^{H-1} c(Z_{t+k+1}, u_{t+k}) \right].$$
(15)

Unlike deterministic JEPA, VJEPA allows for *distributional planning*. We approximate the expectation via Monte Carlo rollouts sampled from the predictor $p_\phi$, enabling the agent to account for aleatoric uncertainty (e.g., via MPPI (Williams et al., 2015)) rather than collapsing to a mean prediction that may be physically invalid. The VJEPA-MPC algorithm is presented in Appendix.G.3.

**Optimality of MAP Planning.** While sampling is general, point estimates are sufficient under specific conditions.

**Theorem 5.3** (Optimality of MAP Control). *If the predictive model is Gaussian with action-independent covariance $\Sigma$ and the cost is quadratic, then minimizing the expected cost is equivalent to minimizing the cost of the mean (MAP) trajectory:*

$$\arg\min_u \mathbb{E}[c(Z_{t+1}, u_t)] = \arg\min_u c(\mu_\phi(Z_t, u_t), u_t).$$

(Proof see Appendix.G.2). This justifies the use of deterministic planning methods in regimes where uncertainty is homoscedastic.

## 6. Bayesian JEPA (BJEPA)

While VJEPA models predictive uncertainty, it implicitly entangles environmental dynamics with the specific distribution of the training data. We introduce *Bayesian JEPA*

*(BJEPA)*, a modular extension that explicitly factorizes the predictive belief into two independent components: a learned *Likelihood Expert* (encoding physics) and a modular *Prior Expert* (encoding constraints or goals).

### 6.1. Product of Experts Factorization

We approximate the posterior distribution of the future representation $Z_T$, given context $Z_C$ and auxiliary task information $\eta$, via a Product of Experts (PoE): small

$$p(Z_T \mid Z_C, \eta) \propto \underbrace{p_{\text{like}}(Z_T \mid Z_C)}_{\text{Dynamics Expert}} \times \underbrace{p_{\text{prior}}(Z_T \mid \eta)}_{\text{Constraint Expert}}.$$
(16)

This Bayesian posterior-like factorization, as detailed in Appendix.J, implements a logical AND operation in probability space: a predicted state has high probability only if it is both *physically feasible* (consistent with history) and *task-compliant* (consistent with constraints). Note that we explicitly maximize the log-likelihood of the *dynamics expert* $p_{\text{like}}(Z_T \mid Z_C)$ rather than the full posterior during the pre-training phase. This factorization is deliberate: it forces the likelihood expert to learn objective physics, preventing it from overfitting to specific task constraints (e.g., stationarity) that might be imposed by the prior.

### 6.2. Architecture

BJEPA builds on the standard JEPA/VJEPA architecture by adding an explicit latent-space prior encoder. The full architecture is visualized in Figure 2. Specifically, a context encoder $f_\theta$ maps historical observations $x_C$ to a latent context $Z_C$, while a target encoder $f_{\theta'}$ provides stable regression targets $Z_T$ from future observations $x_T$ during training. The predictive model is split into two experts: a *predictive likelihood* $p_{\text{like}}(Z_T \mid Z_C)$, parameterized by a dynamics network, which captures physical transitions; and a *latent prior* $p_{\text{prior}}(Z_T \mid \eta)$, parameterized by a prior network, which encodes goals or constraints $\eta$. The final 'Bayesian posterior' $p(Z_T \mid Z_C, \eta)$ is derived as the product of these likelihood and prior distributions.

### 6.3. Training: Structural Regularization

BJEPA exploits the prior during training to filter out nuisance variability. By setting $p_{\text{prior}}$ to a *structural prior* $p_{\text{struct}}$ (e.g., stationarity or slowness), we penalize the dynamics model for tracking high-entropy, unpredictable details. The objective extends Eq. 8 with a structural KL term:

$$\mathcal{L}_{\text{BJEPA}} = \mathcal{L}_{\text{VJEPA}} + \gamma \mathbb{E} \left[ \text{KL}(p_{\text{like}}(Z_T | Z_C) \,\|\, p_{\text{struct}}(Z_T)) \right].$$
(17)

As we show in experiments (Section 7), this regularization allows BJEPA to ignore "Noisy TV" distractors (see later experiment) that violate the stationarity assumption.

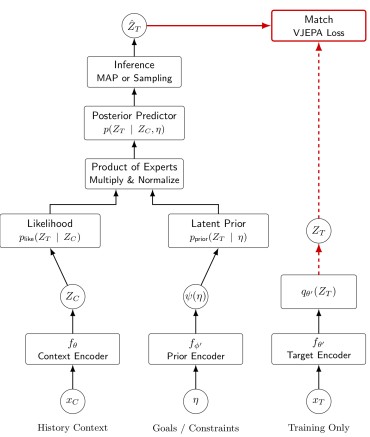

*Figure 2.* BJEPA Architecture. Dynamics (Likelihood) and Constraints (Prior) are fused via PoE to predict $Z_T$. Training matches the PoE output or Likelihood output against the target encoder.

### 6.4. Inference: Zero-Shot Transfer via Prior Swapping

A key advantage of BJEPA is modularity: the dynamics expert can be reused while the prior expert is swapped at inference time. We consider two types of priors: *Learned goal priors* define $p_{\text{prior}}(Z_T \mid x_g) = \mathcal{N}(Z_T; f_{\bar{\theta}}(x_g), \Sigma_g)$ for a goal observation $x_g$, while *Energy-based priors* define $p_{\text{prior}}(Z_T \mid \eta) \propto \exp\{-E_\eta(Z_T)\}$ to impose analytic constraints such as safety or feasibility. The PoE then acts as a Bayesian-style latent update, combining physically plausible predictions with goal- or constraint-compatible priors (Särkkä, 2013). We validate learned goal-prior swapping in Appendix.K.4.

### 6.5. Comparison with Existing Frameworks

Table 2 contextualizes BJEPA within the landscape of latent world models. BJEPA uniquely combines likelihood-free learning (avoiding pixel reconstruction) with explicit uncertainty modeling and modular task specification.

*Table 2.* Comparison of latent world modeling frameworks.

| Method | Obs. Recon. | Uncert. | Task Spec. | Reward |
|---|---|---|---|---|
| Dreamer | ✓ | ✓ | Monolithic | Learned |
| JEPA-WM | ✗ | ✗ | Monolithic | Learned |
| VJEPA | ✗ | ✓ | Implicit | None |
| **BJEPA** | ✗ | ✓ | **Modular** | **Prior/ Energy** |

## 7. Experiments: The Noisy TV Problem

We evaluate[1] whether predictive latent objectives are more robust than reconstruction-based objectives when a low-variance task-relevant signal is mixed with high-variance,

[1]Code available at this Github repository.

unpredictable nuisance variation. The central question is: can a model identify and predict the low-dimensional signal when observations are dominated by a "Noisy TV" distractor? We report the main controlled linear-Gaussian experiment in this section and summarize additional nonlinear, image-based, natural-image, zero-shot prior-conditioning, and control experiments at the end. Experimental details and further results are provided in appendices.

### 7.1. Experimental Setup

We construct a linear-Gaussian system in which the observation $x_t \in \mathbb{R}^{20}$ is a superposition of a task-relevant signal $s_t$ and a nuisance distractor $d_t$:

$$x_t = Cs_t + D(\sigma d_t) + \epsilon_t. \tag{18}$$

Here $C$ and $D$ are fixed random mixing matrices, $\epsilon_t$ denotes small observation noise, and $\sigma$ controls the distractor strength.

- **Signal ($s_t$):** A low-variance latent state evolving under stable linear dynamics with dimension $D_s = 4$. This represents the predictable structure relevant for downstream control or forecasting.

- **Distractor ($d_t$):** A high-variance nuisance process with dimension $D_d = 4$. We vary the distractor scale $\sigma \in \{0, 4, 8\}$. At $\sigma = 8$, the distractor variance is roughly $64\times$ larger than at unit scale, yielding a low-SNR setting.

We compare JEPA, VJEPA, and BJEPA against two reconstruction-based baselines: a **VAE**, trained with an ELBO objective, and an autoregressive **AR** model, trained to predict observations. All models use linear backbones with bottleneck dimension equal to the signal dimension, $D_z = D_s = 4$. This creates a hard capacity trade-off: a model cannot faithfully encode both the task-relevant signal and the high-variance distractor. Full implementation details are given in Appendix.K.

### 7.2. Results and Analysis

**Reconstruction-based baselines are sensitive to high-variance distractors.** As discussed in Section 3, reconstruction-based objectives are encouraged to allocate capacity to observation-level variation, including nuisance variation, when it dominates the reconstruction loss. This behavior appears clearly in the high-noise regime. As shown in Figure 3 and Table 3, the VAE achieves near-perfect signal recovery at $\sigma = 0$, but degrades substantially at $\sigma = 8$ with mean signal $R^2 = -0.226 \pm 0.846$ and noise $R^2 = 0.573 \pm 0.265$. The AR baseline shows a similar trend, reaching only $0.118 \pm 0.352$ signal $R^2$ at $\sigma = 8$. These

results are consistent with the hypothesis that observation-space prediction and reconstruction objectives can spend bottleneck capacity on high-variance nuisance factors.

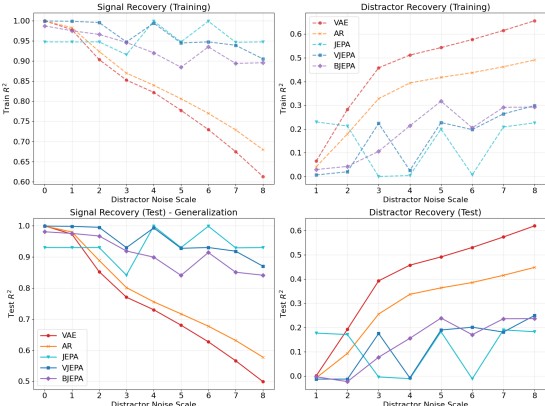

*Figure 3.* **Signal $R^2$ vs. distractor scale.** Reconstruction-based baselines degrade as nuisance variance increases, while predictive JEPA-family objectives retain stronger signal recovery in this controlled setting.

**Predictive objectives improve nuisance robustness.** Across five independent random environments, JEPA-family models generally outperform reconstruction-based baselines in signal recovery. At moderate noise, $\sigma = 4$, VJEPA achieves the best mean signal recovery, with signal $R^2 = 0.796 \pm 0.187$, and the lowest mean noise recovery, with noise $R^2 = 0.129 \pm 0.151$. This is consistent with the predictive-objective interpretation: the model is rewarded for preserving information useful for predicting future or target latents, rather than for reconstructing all observation-level variability.

At high noise, $\sigma = 8$, deterministic JEPA is the most stable model in this linear experiment, achieving signal $R^2 = 0.749 \pm 0.084$ and the lowest noise $R^2 = 0.203 \pm 0.052$. VJEPA remains substantially better than the VAE and AR baselines, but shows higher seed-to-seed variability, with signal $R^2 = 0.566 \pm 0.224$. BJEPA underperforms in this extreme-noise linear setting, suggesting that the structural prior and optimization details can be important in finite-capacity regimes. These results support the claim that predictive objectives *permit* nuisance discarding, but they do not imply that every probabilistic or Bayesian extension will dominate deterministic JEPA under every training condition. The large standard deviations at $\sigma = 8$ also show that single-seed reporting is insufficient in this regime.

**Predictive variance separates signal and nuisance dimensions.** Figure 5 visualizes the mean predicted variance per latent dimension learned by VJEPA at three distractor scales. At $\sigma = 0$, all latent dimensions show low and relatively uniform predictive variance. As the distractor scale increases, the learned predictive variance becomes more

*Table 3.* Multi-seed ($N$=5) Noisy TV results on the test set, reported as mean±std. ↑Sig. $R^2$ higher is better; ↓Noi. $R^2$ lower is better; "–" indicates that noise recovery is undefined at Scale 0; bold indicates the best mean in each block.

| Scale (SNR) | Model | ↑ Sig. $R^2$ | ↓ Noi. $R^2$ |
|---|---|---|---|
| 0.0 ($\infty$ dB) | VAE | $1.000 \pm 0.000$ | – |
| | AR | $0.999 \pm 0.001$ | – |
| | JEPA | $0.999 \pm 0.001$ | – |
| | VJEPA | $0.999 \pm 0.001$ | – |
| | BJEPA | $0.945 \pm 0.074$ | – |
| 4.0 (3.8 dB) | VAE | $0.605 \pm 0.110$ | $0.293 \pm 0.168$ |
| | AR | $0.665 \pm 0.111$ | $0.204 \pm 0.128$ |
| | JEPA | $0.717 \pm 0.259$ | $0.139 \pm 0.164$ |
| | VJEPA | $\mathbf{0.796 \pm 0.187}$ | $\mathbf{0.129 \pm 0.151}$ |
| | BJEPA | $0.750 \pm 0.137$ | $0.170 \pm 0.143$ |
| 8.0 ($-2.2$ dB) | VAE | $-0.226 \pm 0.846$ | $0.573 \pm 0.265$ |
| | AR | $0.118 \pm 0.352$ | $0.414 \pm 0.217$ |
| | JEPA | $\mathbf{0.749 \pm 0.084}$ | $\mathbf{0.203 \pm 0.052}$ |
| | VJEPA | $0.566 \pm 0.224$ | $0.240 \pm 0.105$ |
| | BJEPA | $0.077 \pm 0.340$ | $0.327 \pm 0.115$ |

heterogeneous: dimensions aligned with predictable signal tend to retain low variance, while dimensions exposed to unpredictable nuisance variation exhibit larger predictive variance. This behavior is consistent with the nuisance-invariance analysis: the predictive objective is not forced to explain unpredictable nuisance variation with a confident point prediction, and can instead express uncertainty in those directions.

### 7.3. Further Experiments on Nuisance Robustness

We further test whether the same qualitative pattern extends beyond the linear-Gaussian setting. Full protocols, figures, and tables are provided in Appendix Sections K.2 to K.4, K.9 and K.10.

**Experiment 2: Nonlinear backbone (Appendix.K.2).** Replacing each linear layer with a two-layer Tanh MLP preserves the main trend. At $\sigma = 8$ (SNR $-2.2$ dB), JEPA achieves signal $R^2 = 0.926$ and VJEPA achieves $0.850$, compared with $0.493$ for VAE and $0.567$ for AR. This suggests that the nuisance-robustness effect is not merely an artifact of the linear-Gaussian basis.

**Experiment 3: Temporal MNIST (Appendix.K.3).** On MNIST sequences with random image distractors, VJEPA achieves $61.6\%$ next-digit accuracy at $\sigma = 0$, compared with $38.0\%$ for JEPA. In this setting, explicit predictive uncertainty appears beneficial even when no high-variance distractor is present. Across the tested distractor scales, the ordering VJEPA > BJEPA > JEPA is observed.

**Experiment 4: BJEPA zero-shot goal conditioning (Appendix.K.4).** BJEPA enables prior swapping at inference time. Replacing the stationarity prior with a class-goal embedding, without retraining the dynamics model, increases digit-class accuracy from $56.2\%$ to $100.0\%$ at goal variance $\sigma_{\text{goal}}^2 = 0.30$. The wrong-goal condition drops to $8.3\%$, near chance. This supports the intended modularity of the

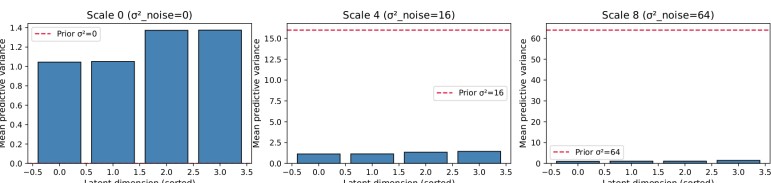

*Figure 4.* **Representative latent reconstructions at $\sigma = 8.0$.** Reconstruction-based baselines track more nuisance variation, while predictive latent objectives more closely follow the task-relevant signal. Aggregate multi-seed results are reported in Table 3.

*Figure 5.* VJEPA per-dimension predictive variance at 3 distractor scales. As the distractor scale increases, some latent dimensions retain low predictive variance while others express high uncertainty, suggesting separation between predictable signal and unpredictable nuisance variation.

Product-of-Experts formulation, although broader validation is needed to establish robustness across tasks.

**Experiment 7: ViT encoder on STL-10 (Appendix.K.9).** A ViT-Tiny (Dosovitskiy et al., 2021) trained from scratch on STL-10 natural images (Coates et al., 2011) reproduces the qualitative ordering BJEPA $\geq$ VJEPA $>$ JEPA. At $\sigma = 1.0$, BJEPA-ViT achieves $17.2\%$ accuracy versus $11.3\%$ for JEPA-ViT, with chance at $10\%$. Although absolute accuracies remain modest, this result suggests that the effect is not restricted to linear or MLP backbones.

**Experiment 8: VJEPA-MPC on DM Control (Appendix.K.10).** On DMC Cheetah-run, VJEPA-MPC achieves a mean return of 9.5 vs. 7.8 (JEPA-MPC) and $\approx 2.3$ (Dreamer-lite, single seed). Dreamer-lite's reward-model loss remains close to $0.998$, consistent with the pixel-ELBO generative collapse observed on Noisy-TV.

## 8. Discussion

**JEPA as probabilistic latent prediction.** We frame JEPA-style learning as predictive latent-state modeling: predicting future representations rather than reconstructing observations. Reconstruction-based objectives (e.g., Dreamer (Hafner et al., 2020)) behave like nonlinear PCA, allocating capacity to dominant directions of observation variance; predictive latent objectives behave more like nonlinear CCA, preserving information shared between context and future latents. The VJEPA latent NLL optimizes a Barber–Agakov lower bound on predictive mutual information, explaining why predictive objectives can discard high-variance nuisance variation that is not useful for predicting future latent structure.

**Practical stability and baselines.** In high-capacity CNN

settings, VICReg regularization was needed to prevent trivial high-uncertainty solutions and unstable latent dynamics (Appendix Sections K.5 and K.7). The RSSM (Appendix.K.8) and Dreamer-lite (Appendix.K.10) baselines exhibit this tension: reconstruction ELBOs fit nuisance-specific information (negative held-out $R^2$; reward-model loss $\approx 0.998$) that does not generalize to reward prediction.

**Limitations.** First, the Gaussian head is unimodal; a preliminary $K=2$ GMM variant (Appendix.I) shows promise but requires multi-seed validation (Huang, 2025). Second, VJEPA's advantage is not uniform: at Scale 8.0, JEPA achieves lower signal-$R^2$ variance than VJEPA (0.084 vs. 0.224), reflecting sensitivity to optimization. Third, Exps. 7–8 are single-seed; multi-seed RL evaluation and stronger encoders remain future work.

## 9. Conclusion

We introduced **VJEPA**, a probabilistic extension of JEPA that learns predictive distributions over future latents without pixel-level likelihoods, and **BJEPA**, a modular Product-of-Experts extension which enables constraint-aware prediction and zero-shot prior swapping. The framework gives JEPA explicit uncertainty semantics, connects latent representation prediction to predictive information, Bayesian filtering, and sufficient information states for control, while preserving the reconstruction-free principle of JEPA. Across Noisy-TV, nonlinear, image-based, ViT/STL-10, and DMC Cheetah-run experiments, JEPA-family predictive objectives are consistently more robust to high-variance nuisance distractors than reconstruction-based baselines. These results support probabilistic latent prediction as a principled route towards robust, uncertainty-aware, reconstruction-free world models for planning and representation learning.

## Impact Statement

This paper advances probabilistic self-supervised world models for representation learning and planning. By learning predictive latent distributions without pixel-level reconstruction, the proposed framework may improve robustness and uncertainty estimation in settings with high-dimensional observations, nuisance variation, or stochastic dynamics. Potential applications include robotics, autonomous systems, infrastructure monitoring, healthcare operations, and other sequential decision-making problems where calibrated uncertainty and constraint-aware planning are important.

At the same time, such models should be deployed cautiously in safety-critical settings. Misspecified priors, distribution shift, or poorly calibrated uncertainty estimates could lead to unreliable plans or overconfident decisions. The present work is therefore best viewed as a methodological contribution requiring further validation before real-world deployment. We do not identify additional societal impacts specific to this work beyond the broader impacts of advances in machine learning and model-based decision-making.

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

| Feature | JEPA | VJEPA |
|---|---|---|
| Target prediction | $\hat{Z}_T = g_\phi(Z_C, \xi_T)$ | $p_\phi(Z_T \mid Z_C, \xi_T)$ |
| Prediction type | Point estimate | Predictive distribution |
| Training objective | MSE or fixed-variance Gaussian NLL | Latent NLL + target-side KL regularization |
| Uncertainty modeling | Implicit fixed variance | Explicit learned variance / latent belief |
| Multi-modal futures | Not represented directly | Possible through particles, ensembles, or richer heads[†] |
| Collapse control | EMA, asymmetry, and predictor design | Predictive mismatch, target regularization, etc. |
| Probabilistic interpretation | Auxiliary fixed-variance Gaussian view | Explicit latent predictive model |
| Compatibility with filtering/control | Limited uncertainty representation | Natural latent belief propagation |

*Table 4.* Comparison between deterministic JEPA and VJEPA. VJEPA preserves the representation-prediction principle of JEPA while adding an explicit latent predictive distribution. [†]The current diagonal-Gaussian implementation is unimodal at each prediction step; multi-modal beliefs require particle rollouts, ensembles, or richer predictive families such as GMMs.

## A. JEPA vs. VJEPA Feature Comparison

## B. VJEPA Training Algorithm

---

**Algorithm 1** VJEPA Training

---

**Require:** Dataset $\{x^{(i)}\}_{i=1}^N$, context encoder $f_\theta$, target encoder $f_{\bar{\theta}}$, predictive model $p_\phi$, prior $p_0$, EMA rate $\tau$, regularization weight $\beta$

1: **for** each minibatch $\{x\}$ **do**
2:      Sample context-target pairs $(x_C, x_T)$ and corresponding side information $\xi_T$
3:      Encode context: $Z_C \leftarrow f_\theta(x_C)$
4:      Compute target distribution: $q_{\bar{\theta}}(Z_T \mid x_T)$
5:      Compute predictive distribution: $p_\phi(Z_T \mid Z_C, \xi_T)$
6:      Compute latent predictive NLL: $\mathcal{L}_{\text{pred}} = \mathbb{E}_{Z_T \sim q_{\bar{\theta}}(\cdot \mid x_T)}[-\log p_\phi(Z_T \mid Z_C, \xi_T)]$
7:      Compute target-side KL regularizer: $\mathcal{L}_{\text{KL}} = \text{KL}(q_{\bar{\theta}}(Z_T \mid x_T) \,\|\, p_0(Z_T))$
8:      Minimize $\mathcal{L} = \mathcal{L}_{\text{pred}} + \beta \mathcal{L}_{\text{KL}}$ with respect to the trainable online parameters and predictor parameters
9:      Update target encoder via EMA: $\bar{\theta} \leftarrow \tau\bar{\theta} + (1-\tau)\theta$
10: **end for**

---

## C. Equivalence Between Squared Loss and Gaussian Likelihood

In Section 3, we stated that minimizing the deterministic JEPA regression loss is equivalent to maximizing the log-likelihood under an isotropic Gaussian predictive model.

Consider the probabilistic model:

$$p(Z_T \mid Z_C) = \mathcal{N}\left(Z_T \mid \hat{Z}_T, \sigma^2 I\right), \quad \text{where } \hat{Z}_T = g_\phi(Z_C, \xi_T).$$

The negative log-likelihood (NLL) is:

$$-\log p(Z_T \mid Z_C) = \frac{d}{2}\log(2\pi\sigma^2) + \frac{1}{2\sigma^2}\|Z_T - \hat{Z}_T\|^2.$$

Since $\sigma^2$ is fixed in the deterministic formulation, minimizing the NLL is equivalent to minimizing the squared error $\|Z_T - \hat{Z}_T\|^2$. VJEPA generalizes this by making $\sigma^2$ (or the full covariance $\Sigma$) a learnable output of the network, allowing the model to down-weight uncertain predictions.

## D. Information-Theoretic Identities

We use the standard conditional entropy and mutual-information identity

$$H(A \mid C) - H(A \mid B, C) = I(A; B \mid C),$$

which is applied in the collapse and nuisance-invariance arguments below.

# E. Proof of Collapse Avoidance

We provide the formal argument that the VJEPA objective prevents representation collapse under the assumption of target diversity.

**Theorem E.1** (No Collapsed Global Optimum under Target Diversity). *Assume (i) Target Diversity: there exist $x_T, x'_T$ such that $q_{\theta'}(\cdot|x_T) \neq q_{\theta'}(\cdot|x'_T)$, and (ii) Nontrivial Conditioning. Then no global minimizer of $\mathcal{L}_{VJEPA}$ satisfies $f_\theta(x_C) \equiv c$.*

*Proof.* Assume the context encoder collapses to a constant $c$. The predictor becomes an unconditional distribution $p_\phi(Z_T \mid c, \xi_T)$. For a fixed $\xi_T$, the optimal unconditional predictor minimizes the cross-entropy with the aggregated target distribution $\bar{q}(Z_T) = \mathbb{E}_{x_T}[q_{\theta'}(Z_T|x_T)]$. The minimum achievable loss is the entropy of the mixture:

$$\mathcal{L}_{\text{collapsed}} = H(Z_T \mid \xi_T) = H(Z_T \mid X_T, \xi_T) + I(Z_T; X_T \mid \xi_T).$$

The first term is the expected uncertainty of the target encoder (aleatoric). The second term is the mutual information between the target embedding and the target input. Under the assumption of Target Diversity, $I(Z_T; X_T \mid \xi_T) > 0$. A non-collapsed encoder can condition on $Z_C$ to approximate the specific posterior $q_{\theta'}(Z_T|x_T)$, achieving a loss closer to $H(Z_T \mid X_T, \xi_T)$. The difference $\mathcal{L}_{\text{collapsed}} - \mathcal{L}_{\text{optimal}} \approx I(Z_T; X_T \mid \xi_T) > 0$. Thus, the collapsed solution is strictly suboptimal.

**Detailed proof**   The proof follows three steps: we first analyze the structural consequences of a collapsed context encoder, then derive the minimal achievable prediction loss in this regime, and finally demonstrate that non-collapsed representations achieve strictly lower VJEPA objective values.

**Step 1: Consequence of a collapsed context encoder.**   Assume, for contradiction, that the context encoder collapses:

$$f_\theta(x_C) = c \qquad \forall x_C.$$

Then the predictive model cannot depend on the context content and reduces to

$$p_\phi(Z_T \mid Z_C, \xi_T) = p_\phi(Z_T \mid c, \xi_T),$$

i.e. an *unconditional* distribution for each $\xi_T$. Under this restriction, the first term of the VJEPA objective becomes

$$\mathcal{L}_{\text{pred}}^{\text{coll}} = \mathbb{E}_{\xi_T} \mathbb{E}_{x_T|\xi_T} \mathbb{E}_{Z_T \sim q_{\theta'}(\cdot|x_T)} \big[ - \log p_\phi(Z_T \mid c, \xi_T) \big]. \tag{19}$$

**Step 2: Optimal unconditional predictor.**   For a fixed $\xi_T$, define the *aggregated target distribution*

$$\bar{q}(Z_T \mid \xi_T) := \mathbb{E}_{x_T|\xi_T} \big[ q_{\theta'}(Z_T \mid x_T) \big].$$

It is a standard result that the distribution minimizing cross-entropy with respect to $\bar{q}$ is $\bar{q}$ itself. Hence,

$$\inf_{p(\cdot|c,\xi_T)} \mathbb{E}_{x_T|\xi_T} \mathbb{E}_{Z_T \sim q_{\theta'}(\cdot|x_T)}[- \log p(Z_T)] = H(\bar{q}(\cdot \mid \xi_T)).$$

Therefore, the best achievable collapsed prediction loss equals

$$\mathcal{L}_{\text{pred}}^{\text{coll}} = \mathbb{E}_{\xi_T} H(Z_T \mid \xi_T). \tag{20}$$

**Step 3: Decomposition via conditional mutual information.**   By standard entropy identities (see Appendix D),

$$H(Z_T \mid \xi_T) = H(Z_T \mid X_T, \xi_T) + I(Z_T; X_T \mid \xi_T).$$

Assumption (i) (*target diversity*) implies that

$$I(Z_T; X_T \mid \xi_T) > 0$$

for at least one $\xi_T$ with nonzero probability mass. Hence the collapsed predictor incurs a strictly positive irreducible excess loss.

**Step 4: Advantage of non-collapsed representations.** Consider any non-collapsed encoder producing distinct $Z_C$ values correlated with $x_T$. By Assumption (ii), the predictive family can represent different $p_\phi(Z_T \mid Z_C, \xi_T)$ and therefore approximate $q_{\theta'}(Z_T \mid x_T)$ conditionally. In the realizable limit,

$$\mathcal{L}_{\text{pred}}^{\text{non-coll}} = \mathbb{E}_{\xi_T} \mathbb{E}_{x_T \mid \xi_T} H(Z_T \mid X_T = x_T, \xi_T) < \mathcal{L}_{\text{pred}}^{\text{coll}}.$$

**Step 5: Role of the KL regularizer.** The KL term

$$\text{KL}(q_{\theta'}(Z_T \mid x_T) \,\|\, p(Z_T))$$

is independent of $Z_C$ and therefore does not eliminate the gap identified above. Its role is to prevent pathological collapse of the target encoder, not to create the strict separation between collapsed and non-collapsed optima.

**Conclusion** A collapsed context encoder forces the predictive model to fit a single unconditional distribution to multiple distinct target distributions, incurring an irreducible mutual-information gap. Since a non-collapsed solution can achieve strictly lower loss, no collapsed representation can be globally optimal.

**Interpretation.** Collapse is avoided because predictive uncertainty depends on the target input. VJEPA therefore prevents collapse through *information mismatch*, not architectural heuristics. Theorem 3.2 formalizes the intuitive point that if target embeddings vary with the target input, then a collapsed context representation forces the predictor to fit a single unconditional distribution to multiple distinct targets, incurring an irreducible prediction loss. Consequently, any solution satisfying the collapsed mode hypothesis cannot be globally optimal for the VJEPA objective. Under the stated assumptions, preventing collapse is therefore an intrinsic property of the objective itself, rather than a consequence of architectural heuristics or training asymmetries. $\square$

## F. Proof of Nuisance Invariance

**Proposition.** *Let observation $x = S + N$, where $S$ is a predictable signal and $N$ is high-entropy nuisance noise independent of the future target $Z_{t+\Delta}$. Generative models maximizing $p(x|Z)$ must encode $N$. VJEPA is invariant to representations that discard $N$.*

*Proof.* Let the observation be $x_t = (s_t, n_t)$. We assume the future target $Z_{t+\Delta}$ depends only on the signal $s_t$, implying $I(n_t; Z_{t+\Delta} \mid s_t) = 0$.

**Case 1: Generative Objective.** A generative model maximizes $\mathcal{L}_{\text{gen}} = \mathbb{E}[\log p(x_t \mid Z_t)]$. Using the chain rule, $\log p(s_t, n_t \mid Z_t) = \log p(s_t \mid Z_t) + \log p(n_t \mid s_t, Z_t)$. To maximize this objective, $Z_t$ must minimize the conditional entropy $H(n_t \mid s_t, Z_t)$. This forces $Z_t$ to encode information about $n_t$. A minimal representation $Z_t = s_t$ is strictly suboptimal because it fails to reconstruct $n_t$.

**Case 2: VJEPA Objective.** VJEPA maximizes the predictive likelihood $\mathcal{L}_{\text{VJEPA}} = \mathbb{E}[\log p(Z_{t+\Delta} \mid Z_t)]$. Since $Z_{t+\Delta}$ is conditionally independent of $n_t$ given $s_t$, the true predictive distribution satisfies $p(Z_{t+\Delta} \mid s_t, n_t) = p(Z_{t+\Delta} \mid s_t)$. Therefore, a minimal representation $Z_t^{\min} = s_t$ achieves the same optimal likelihood as a maximal representation $Z_t^{\max} = (s_t, n_t)$. Since the objective does not penalize the absence of $n_t$, the optimization landscape admits solutions that filter out nuisance variability. $\square$

## G. Control Theory Proofs

### G.1. Proof of Optimal Control Sufficiency

*Statement: If $Z_t$ is predictively sufficient $(p(Z_{t+1:H}|h_t, u) = p(Z_{t+1:H}|Z_t, u))$, there exists an optimal policy $\pi^\star(u_t|Z_t)$.*

*Proof.* This follows from the standard sufficient statistic argument in optimal control. Let $J(h_t)$ be the optimal value function given history $h_t$. The value function satisfies the Bellman equation:

$$J(h_t) = \min_{u_t} \mathbb{E}\left[c(Z_{t+1}, u_t) + J(h_{t+1}) \mid h_t\right].$$

By predictive sufficiency, the distribution of $Z_{t+1}$ and all future costs depends only on $Z_t$ and the action sequence. Therefore, we can define a reduced value function $V(Z_t)$ such that $V(Z_t) = J(h_t)$ for all histories mapping to $Z_t$. The optimal policy is the argmin of the Bellman operator on $V(Z_t)$, which depends only on $Z_t$. $\qquad\square$

### G.2. Optimality of MAP Planning

We prove that planning using the mean/MAP trajectory is sufficient under specific conditions (Linear-Gaussian Dynamics, Quadratic Cost).

**Theorem G.1.** *Let $p_\phi(Z_{t+1}|Z_t, u_t) = \mathcal{N}(\mu_\phi(Z_t, u_t), \Sigma)$ with action-independent covariance $\Sigma$. Let cost be $c(Z, u) = (Z - g)^\top Q(Z - g) + r(u)$. Then $\arg\min_u \mathbb{E}[c(Z_{t+1}, u_t)]$ is equivalent to $\arg\min_u c(\mu_\phi(Z_t, u_t), u_t)$.*

*Proof.* Expand the expected quadratic cost for $Z \sim \mathcal{N}(\mu, \Sigma)$:

$$\mathbb{E}[(Z - g)^\top Q(Z - g)] = (\mu - g)^\top Q(\mu - g) + \text{tr}(Q\Sigma).$$

Since $\Sigma$ is independent of $u_t$, the trace term is constant w.r.t optimization. Minimizing the expected cost is equivalent to minimizing the cost evaluated at the mean $\mu$. For Gaussian distributions, the mean is the MAP estimate.

**Detailed proof**   We want to analyze this objective

$$J(u_t) := \mathbb{E}\left[c(Z, u_t) \mid Z_t, u_t\right] = \mathbb{E}\left[(Z - z^\star)^\top Q_c(Z - z^\star)\right] + r(u_t). \tag{21}$$

in which the only nontrivial term is the expectation of the quadratic form.

**Step 1: Expand the quadratic form.**   Let $d := Z - z^\star$. Then

$$d^\top Q_c d = (Z - z^\star)^\top Q_c(Z - z^\star).$$

Insert and subtract the mean $\mu$:

$$Z - z^\star = (Z - \mu) + (\mu - z^\star).$$

Therefore,

$$(Z - z^\star)^\top Q_c(Z - z^\star) = \left((Z - \mu) + (\mu - z^\star)\right)^\top Q_c\left((Z - \mu) + (\mu - z^\star)\right).$$

Expand the product into four terms:

$$
\begin{aligned}
\left((Z - \mu) &+ (\mu - z^\star)\right)^\top Q_c\left((Z - \mu) + (\mu - z^\star)\right) \\
&= (\boldsymbol{Z - \mu})^\top \boldsymbol{Q_c}(\boldsymbol{Z - \mu}) + (Z - \mu)^\top Q_c(\mu - z^\star) + (\mu - z^\star)^\top Q_c(Z - \mu) + (\boldsymbol{\mu - z^\star})^\top \boldsymbol{Q_c}(\boldsymbol{\mu - z^\star}).
\end{aligned}
$$

**Step 2: Take expectations term-by-term.**   Take $\mathbb{E}[\cdot]$ under $Z \sim \mathcal{N}(\mu, \Sigma)$.

*(i) Cross terms vanish.* Because $\mathbb{E}[Z - \mu] = 0$, we have

$$\mathbb{E}\left[(Z - \mu)^\top Q_c(\mu - z^\star)\right] = \mathbb{E}[Z - \mu]^\top Q_c(\mu - z^\star) = 0.$$

Similarly,

$$\mathbb{E}\left[(\mu - z^\star)^\top Q_c(Z - \mu)\right] = (\mu - z^\star)^\top Q_c\mathbb{E}[Z - \mu] = 0.$$

*(ii) The constant term remains.* Since $(\mu - z^\star)^\top Q_c(\mu - z^\star)$ is deterministic given $u_t$,

$$\mathbb{E}\left[(\mu - z^\star)^\top Q_c(\mu - z^\star)\right] = (\mu - z^\star)^\top Q_c(\mu - z^\star).$$

*(iii) The centered quadratic term becomes a trace.* Let $\varepsilon := Z - \mu$. Then $\mathbb{E}[\varepsilon] = 0$ and $\mathbb{E}[\varepsilon\varepsilon^\top] = \Sigma$. We claim

$$\mathbb{E}\left[\varepsilon^\top Q_c\varepsilon\right] = \text{tr}(Q_c\Sigma).$$

To see this, use the identity for any vector $\varepsilon$ and matrix $Q_c$:

$$\varepsilon^\top Q_c \varepsilon = \mathrm{tr}\left(\varepsilon^\top Q_c \varepsilon\right) = \mathrm{tr}\left(Q_c \varepsilon \varepsilon^\top\right),$$

where we used $\mathrm{tr}(a) = a$ for a scalar $a$, and cyclicity of trace [2]. Taking expectations,

$$\mathbb{E}\left[\varepsilon^\top Q_c \varepsilon\right] = \mathbb{E}\left[\mathrm{tr}\left(Q_c \varepsilon \varepsilon^\top\right)\right] = \mathrm{tr}\left(Q_c \mathbb{E}[\varepsilon \varepsilon^\top]\right) = \mathrm{tr}(Q_c \Sigma),$$

where linearity of expectation and linearity of trace justify exchanging $\mathbb{E}$ and $\mathrm{tr}$.

**Step 3: Assemble the expected quadratic cost.** Combining the above,

$$\mathbb{E}\left[(Z - z^\star)^\top Q_c (Z - z^\star)\right] = (\mu - z^\star)^\top Q_c (\mu - z^\star) + \mathrm{tr}(Q_c \Sigma).$$

Hence the full objective in Eq. (21) becomes

$$J(u_t) = (\mu_\phi(Z_t, u_t) - z^\star)^\top Q_c (\mu_\phi(Z_t, u_t) - z^\star) + \mathrm{tr}(Q_c \Sigma) + r(u_t).$$

**Step 4: Use the covariance-independence assumption to reduce the argmin.** By assumption, $\Sigma$ does not depend on $u_t$, and therefore $\mathrm{tr}(Q_c \Sigma)$ is a constant with respect to $u_t$. Adding or subtracting a constant does not change the minimizer, so

$$\arg\min_{u_t} J(u_t) = \arg\min_{u_t} \left[(\mu_\phi(Z_t, u_t) - z^\star)^\top Q_c (\mu_\phi(Z_t, u_t) - z^\star) + r(u_t)\right]$$
$$= \arg\min_{u_t} c(\mu_\phi(Z_t, u_t), u_t),$$

which proves the mean-planning claim.

**Step 5: Mean equals MAP for a Gaussian.** For $Z \sim \mathcal{N}(\mu, \Sigma)$ with $\Sigma \succ 0$, the log-density is

$$\log p(Z) = -\tfrac{1}{2}(Z - \mu)^\top \Sigma^{-1}(Z - \mu) + \mathrm{const},$$

which is strictly concave in $Z$ and uniquely maximized at $Z = \mu$. Thus the MAP point equals the mean:

$$\arg\max_Z p(Z) = \mu.$$

Therefore minimizing cost at the predictive mean is equivalently minimizing cost at the predictive MAP point, i.e. MAP control. □

*Remark* G.2 (Assumptions matter). If $\Sigma = \Sigma(u_t)$ depends on the action, then the term $\mathrm{tr}(Q_c \Sigma(u_t))$ is no longer constant and the optimal action generally depends on predictive uncertainty. Likewise, for non-quadratic costs or non-Gaussian/multimodal $p_\phi$, the expected cost typically cannot be reduced to evaluating a single point estimate.

*Remark* G.3 (Post-action cost indexing). We index the stage cost as $c(Z_{t+1}, u_t)$ to emphasize that cost is incurred *after* applying the action $u_t$ and observing its effect on the system. This convention is standard in stochastic MPC and belief-space control, where actions are optimized based on predicted future states rather than current ones (Mesbah, 2016; Rawlings et al., 2017).

Some reinforcement learning and dynamic programming formulations instead write the stage cost as $c(Z_t, u_t)$ (Sutton & Barto, 2018; Puterman, 2014). The two conventions are equivalent under expectation for Markov dynamics, since

$$\mathbb{E}[c(Z_{t+1}, u_t) \mid Z_t, u_t]$$

can be absorbed into a redefined cost function of $(Z_t, u_t)$. We adopt the post-action form to align with prediction-based control and MPC.

### G.3. Algorithm: VJEPA Sampling-Based Latent MPC

We now formalize the sampling-based planning procedure discussed in Section 5. This procedure, commonly referred to as *random shooting* or *Model Predictive Path Integral* (MPPI) control in the literature (Williams et al., 2015; 2016), approximates the minimization of the expected cost (Eq. (15)) by simulating $N$ parallel latent trajectories using the VJEPA predictive model.

---

[2]The cyclic property of the trace says that $\mathrm{tr}(XY) = \mathrm{tr}(YX)$; in our case $X = \varepsilon^T$ and $Y = Q_c \varepsilon$.

---

**Algorithm 2** VJEPA-Based Model Predictive Control (VJEPA-MPC)

---

**Require:** History $x_{\leq t}$, Horizon $H$, Number of samples $M$, Cost function $c$

1: **Encode:** Estimate current predictive state:

$$Z_t = f_\theta(x_{\leq t})$$

2: **for** sample $i = 1$ to $M$ **do**

3:      Sample candidate action sequence $u_{t:t+H-1}^{(i)} \sim p(u)$

4:      $Z_t^{(i)} \leftarrow Z_t$

5:      **for** step $k = 0$ to $H - 1$ **do**

6:          Sample next latent state (dynamics rollout):

$$Z_{t+k+1}^{(i)} \sim p_\phi(Z_{t+k+1} \mid Z_{t+k}^{(i)}, u_{t+k}^{(i)})$$

7:      **end for**

8:      Compute cumulative cost for trajectory $i$:

$$J^{(i)} = \sum_{k=0}^{H-1} c(Z_{t+k+1}^{(i)}, u_{t+k}^{(i)})$$

9: **end for**

10: **Select:** Optimal action sequence index $i^\star = \arg\min_i J^{(i)}$

11: **Execute:** Apply first action $u_t^{(i^\star)}$ to environment

---

## H. Filtering Comparison Table

The main paper interprets VJEPA as a latent predictive model compatible with Bayesian filtering. The detailed Kalman- and particle-filter analogies are omitted here for brevity; the key empirical comparison against an oracle Kalman filter is retained in Table 5.

*Table 5.* Signal $R^2$ for Oracle KF, JEPA, and VJEPA on the Noisy TV benchmark (seed 111). The oracle KF has full system knowledge; learned methods must identify the structure from data.

| | **Scale 0.0** (SNR 92 dB) | | **Scale 4.0** (SNR 3.8 dB) | | **Scale 8.0** (SNR $-2.2$ dB) | |
| --- | --- | --- | --- | --- | --- | --- |
| **Model** | $R^2$ | Gap | $R^2$ | Gap | $R^2$ | Gap |
| KF (Oracle) | 0.998 | — | 0.998 | — | 0.998 | — |
| JEPA | 0.998 | 0.000 | 0.223 | 0.775 | 0.569 | 0.429 |
| VJEPA | 0.998 | 0.000 | 0.662 | 0.336 | 0.208 | 0.790 |

## I. GMM-VJEPA: Preliminary Results

**Setup.** We replace the diagonal-Gaussian predictor in VJEPA with a $K{=}2$ Gaussian mixture head. The mixture weights are produced by a softmax head, each component has its own mean and diagonal variance, and the predictive NLL is computed using a log-sum-exp mixture likelihood. The target encoder remains Gaussian, so the target-side KL remains tractable. All other settings match the main Noisy-TV experiment.

**Results.** Table 6 reports a single-seed feasibility study. The GMM head improves performance at the hardest noise scale but is unstable at moderate noise, so we treat it as preliminary evidence that richer predictive families are possible rather than as a definitive benchmark.

## J. Derivation of the BJEPA Predictive Factorization

In this section, we derive the relationship between the joint posterior $p(Z_T \mid Z_C, \eta)$ and the individual predictive factors used in the BJEPA architecture (Eq. 16). We aim to express the posterior probability of the target latent state $Z_T$, given

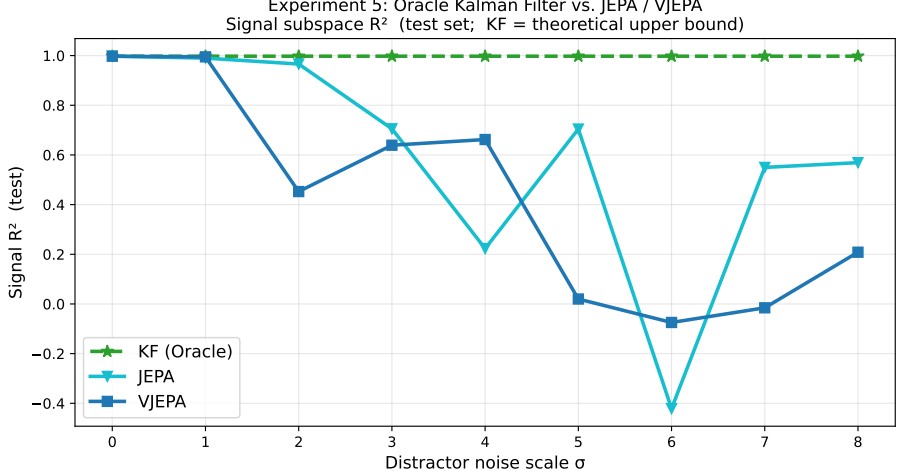

*Figure 6.* Signal $R^2$ vs. distractor scale for Oracle KF (dashed green), JEPA (cyan), and VJEPA (blue). The oracle KF, which knows the true system matrices, maintains near-perfect recovery across all noise levels. Learned methods degrade with increasing distractor variance, with VJEPA outperforming JEPA at moderate noise (Scale 4.0) and JEPA outperforming VJEPA at the highest noise (Scale 8.0), consistent with the single-seed variance observed in Exp. 1.

*Table 6.* Signal $R^2$ and Noise $R^2$ on the linear-Gaussian Noisy TV benchmark (single seed 111). Higher Signal $R^2$ and lower Noise $R^2$ are better. Scale 0.0 Noise $R^2$ is N/A (no distractor variance at $\sigma=0$).

| Model | Scale 0.0 | | Scale 4.0 | | Scale 8.0 | |
|---|---|---|---|---|---|---|
| | Signal $R^2$ | Noise $R^2$ | Signal $R^2$ | Noise $R^2$ | Signal $R^2$ | Noise $R^2$ |
| JEPA | 0.998 | N/A | 0.711 | 0.150 | 0.319 | 0.113 |
| VJEPA | 0.997 | N/A | 0.328 | 0.078 | $-0.502$ | 0.356 |
| GMM-VJEPA | 0.997 | N/A | $-0.244$ | 0.175 | **0.488** | **0.076** |

the historical context $Z_C$ and auxiliary information $\eta$, in terms of the individual conditional probabilities $p(Z_T \mid Z_C)$ and $p(Z_T \mid \eta)$.

**Assumption: Conditional Independence.** We assume that the history encoder and the auxiliary encoder provide independent information about the target state. Formally, we assume that $Z_C$ and $\eta$ are conditionally independent given the true target $Z_T$:

$$p(Z_C, \eta \mid Z_T) = p(Z_C \mid Z_T)\, p(\eta \mid Z_T). \tag{22}$$

This implies that if the true future state is known, the past history and the auxiliary constraints (e.g., goals) do not provide additional information about each other.

**Derivation.** Starting with Bayes' theorem for the posterior:

$$p(Z_T \mid Z_C, \eta) = \frac{p(Z_C, \eta \mid Z_T)\, p(Z_T)}{p(Z_C, \eta)}.$$

Substituting the conditional independence assumption from Eq. (22):

$$p(Z_T \mid Z_C, \eta) = \frac{p(Z_C \mid Z_T)\, p(\eta \mid Z_T)\, p(Z_T)}{p(Z_C, \eta)}.$$

Next, we invert the individual likelihood terms using Bayes' theorem to express them in terms of the predictive distributions (the "experts"):

$$p(Z_C \mid Z_T) = \frac{p(Z_T \mid Z_C)\, p(Z_C)}{p(Z_T)} \quad \text{and} \quad p(\eta \mid Z_T) = \frac{p(Z_T \mid \eta)\, p(\eta)}{p(Z_T)}.$$

Experiment 4: GMM-VJEPA vs. Gaussian VJEPA vs. JEPA
GMM predictor (K=2) retains nuisance-invariance property of VJEPA

*Figure 7.* Signal $R^2$ (left) and Noise $R^2$ (right) across distractor scales for JEPA, VJEPA, and GMM-VJEPA (single seed). At Scale 8.0 (lowest SNR), GMM-VJEPA achieves the highest signal recovery and the lowest noise leakage among the three models, suggesting that distributional flexibility in the predictive head can be beneficial at extreme noise. Performance at Scale 4.0 collapses in this single run, reflecting the high variance already observed for VJEPA in the multi-seed evaluation (Table 3).

Substituting these back into the posterior equation:

$$p(Z_T \mid Z_C, \eta) = \frac{\left(\frac{p(Z_T|Z_C)\,p(Z_C)}{p(Z_T)}\right)\left(\frac{p(Z_T|\eta)\,p(\eta)}{p(Z_T)}\right)p(Z_T)}{p(Z_C, \eta)}.$$

Simplifying the terms:

$$p(Z_T \mid Z_C, \eta) = \frac{p(Z_T \mid Z_C)\,p(Z_T \mid \eta)\,p(Z_C)\,p(\eta)}{p(Z_T)\,p(Z_C, \eta)}.$$

Since $Z_C$ and $\eta$ are observed inputs (constants with respect to the optimization of $Z_T$), the terms $p(Z_C)$, $p(\eta)$, and $p(Z_C, \eta)$ can be absorbed into a normalization constant $\mathcal{Z}$. The relation reduces to:

$$p(Z_T \mid Z_C, \eta) \propto \frac{p(Z_T \mid Z_C)\,p(Z_T \mid \eta)}{p(Z_T)}.$$

In the BJEPA formulation, we approximate the marginal prior over targets $p(Z_T)$ as uniform (or absorbed into the learned bias of the experts), yielding the standard Product of Experts form:

$$p(Z_T \mid Z_C, \eta) \propto p(Z_T \mid Z_C)\,p(Z_T \mid \eta).$$

## K. Experimental Details and Additional Results

This appendix keeps the essential setup and all quantitative result tables for the auxiliary experiments. Unless stated otherwise, experiments compare JEPA-family predictive objectives with reconstruction-based baselines under controlled nuisance variation.

### K.1. Shared Setup and Hyperparameters

The linear Noisy-TV experiments use a controlled linear-Gaussian system with observation dimension $D_x{=}20$, signal dimension $D_s{=}4$, distractor dimension $D_d{=}4$, and bottleneck dimension $D_z{=}4$. The signal follows stable rotational dynamics, the distractor follows an autoregressive nuisance process, and observations are mixed as $x_t = Cs_t + D(\sigma d_t) + \epsilon_t$. Linear Noisy-TV runs were lightweight enough for Colab CPU execution; image and control experiments use GPU acceleration where required. Metrics are signal recovery $R^2$, nuisance/noise recovery $R^2$, probe accuracy from predicted latents, or episode return depending on the benchmark.

Unless otherwise stated, linear experiments use full-batch Adam with learning rate $10^{-3}$, seed 111 for the original run, and seeds $\{42, 111, 123, 456, 789\}$ for multi-seed replication. The baseline models are VAE, AR, JEPA, VJEPA, and BJEPA; JEPA-family methods share the same bottleneck dimension so that models must trade off task-relevant signal against high-variance nuisance variation. Shared optimization and loss settings are summarized in Table 7.

*Table 7.* Optimization settings and loss function hyperparameters.

| Parameter | Value | Description |
|---|---|---|
| *Global Optimization* | | |
| Random Seed | 111 (original); 42/111/123/456/789 (multi-seed) | Original run and replication seeds |
| Optimizer | Adam (Kingma & Ba, 2017) | Standard implementation |
| Learning Rate | $1 \times 10^{-3}$ | Constant throughout training |
| Training Steps | 6000 (original); 3000 (multi-seed) | 3000 sufficient for convergence at $N=5$ |
| *JEPA Family Common Settings* | | |
| EMA Decay ($\tau$) | 0.99 | Target encoder update rate |
| *Model-Specific Coefficients* | | |
| **JEPA** (VICReg Loss) | | |
|    Invariance Coeff | 25.0 | MSE term weight |
|    Variance Coeff | 25.0 | Hinge loss on standard deviation |
|    Covariance Coeff | 1.0 | Off-diagonal decorrelation weight |
| **VJEPA** | | |
|    $\beta$ (Eq. 8) | 0.01 | Target encoder KL regularization weight |
| **BJEPA** | | |
|    $\beta$ (Eq. 8) | 0.01 | Target encoder KL regularization weight |
|    $\gamma$ (Eq. 17) | 0.1 | **Structural Prior** KL weight |

### K.2. Experiment 2: Nonlinear MLP Backbone

This experiment tests whether the Noisy-TV effect is merely an artifact of the linear-Gaussian basis. We replace the linear encoder and predictor with two-layer Tanh MLPs while keeping the same environment, bottleneck dimension, and evaluation protocol. The encoder maps $\mathbb{R}^{20} \rightarrow 32 \rightarrow 4$, the predictor maps $\mathbb{R}^4 \rightarrow 32 \rightarrow 4$, and VJEPA/BJEPA use corresponding mean and variance heads. We train a single seed (111) for 5 000 epochs across all distractor scales $\sigma \in \{0, \ldots, 8\}$.

Table 8 reports selected test-set values and Figure 8 shows all scales. The main pattern from the linear case persists: predictive JEPA-family objectives retain higher signal recovery and lower noise leakage than reconstruction-based baselines at high distractor strength. At $\sigma=8$, JEPA reaches 0.926 signal $R^2$ and VJEPA reaches 0.850, compared with 0.493 for VAE and 0.567 for AR.

*Table 8.* Exp. 2 key results (MLP backbone, seed = 111, single run). Test-set $R^2$.

| Scale (SNR) | Model | ↑ Signal $R^2$ | ↓ Noise $R^2$ |
|---|---|---|---|
| 4.0 (3.8 dB) | VAE | 0.719 | 0.470 |
| | AR | 0.742 | 0.278 |
| | JEPA | **0.994** | **−0.012** |
| | VJEPA | 0.866 | 0.240 |
| | BJEPA | 0.842 | 0.255 |
| 8.0 (−2.2 dB) | VAE | 0.493 | 0.585 |
| | AR | 0.567 | 0.321 |
| | JEPA | **0.926** | **0.173** |
| | VJEPA | 0.850 | 0.243 |
| | BJEPA | 0.082 | 0.374 |

### K.3. Experiment 3: Temporal MNIST with Image Distractors

Temporal MNIST tests whether nuisance robustness extends to real image observations. Each length-$T=8$ sequence cycles through digit classes $0 \rightarrow 1 \rightarrow \cdots \rightarrow 9 \rightarrow 0$. The observation is $x_t = x_t^{\text{signal}} + \sigma x_t^{\text{distractor}}$, clamped to $[0, 1]$, where the distractor is an independently sampled MNIST image and $\sigma \in \{0, 1, 2, 3, 4\}$. Models use an MLP encoder $784 \rightarrow 256 \rightarrow 16$ with Tanh activations. Evaluation uses a logistic-regression probe on the predicted latent to classify the next digit class; MNIST training images generate training sequences and MNIST test images generate held-out evaluation sequences,

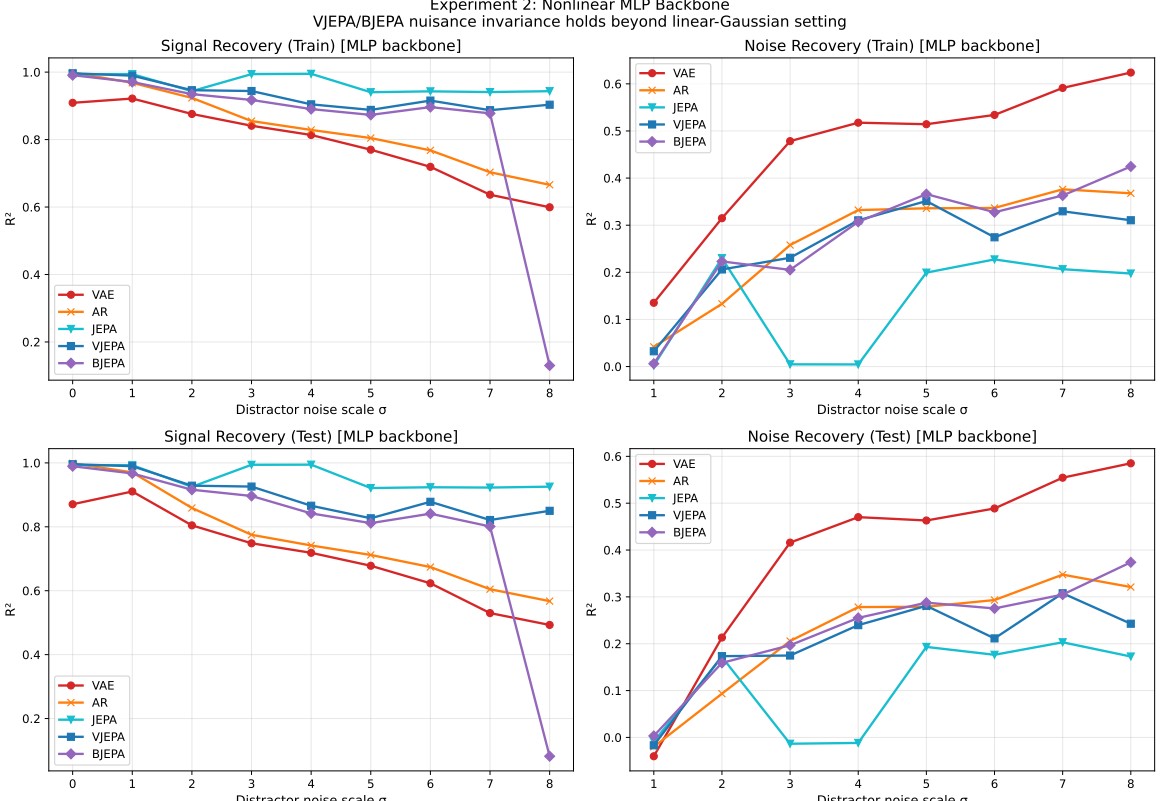

*Figure 8.* Exp. 2: Signal and noise $R^2$ across all distractor scales with MLP encoder/predictor. JEPA-family models (cyan, blue, purple) maintain high signal $R^2$ and low noise $R^2$ throughout. Generative baselines (red, orange) degrade as noise grows. BJEPA collapses at $\sigma=8$ in this single-seed run, consistent with the multi-seed finding.

avoiding leakage.

Table 9 gives next-class test accuracy, while Figure 9 illustrates the signal/distractor construction and Figure 10 plots accuracy across scales. VJEPA is strongest overall, especially at $\sigma=0$ where it obtains $61.6\%$ accuracy versus $38.0\%$ for JEPA, suggesting that explicit latent uncertainty can improve temporal prediction even before strong distractors are added.

*Table 9.* Temporal MNIST: next digit-class test accuracy (%), logistic probe on predicted latents. Chance = 10%. Bold = best per scale.

| Model | $\sigma=0$ | $\sigma=1$ | $\sigma=2$ | $\sigma=3$ | $\sigma=4$ |
|---|---|---|---|---|---|
| JEPA | 38.0 | 13.9 | 12.1 | 11.2 | 10.1 |
| VJEPA | **61.6** | **19.8** | 13.5 | **14.7** | **13.7** |
| BJEPA | 58.7 | 19.4 | **14.4** | 14.0 | 11.7 |
| Chance | 10.0 | 10.0 | 10.0 | 10.0 | 10.0 |

### K.4. Experiment 4: BJEPA Zero-Shot Goal Conditioning

This experiment validates the modular prior-swapping mechanism of BJEPA. We reuse the BJEPA backbone trained in Experiment 3 at $\sigma=0$ and do not retrain the dynamics expert. At inference time, the stationarity prior is replaced by a learned class-goal prior $\mathcal{N}(z_{\text{goal}}, \sigma^2_{\text{goal}}I)$, where $z_{\text{goal}}$ is the mean target-encoder embedding of test images from the desired digit class. We sweep $\sigma^2_{\text{goal}} \in \{0.1, 0.3, 1.0, 3.0, 10.0\}$ and compare no-goal, correct-goal, and wrong-goal conditions.

Table 10 and Figure 11 show that the PoE prior has a controllable effect: tight correct priors drive accuracy to $100\%$, whereas tight wrong priors pull predictions toward the wrong class and drop performance near chance. As the prior variance grows, the model smoothly returns to the no-goal baseline.

Temporal MNIST: Signal + Noisy TV Distractor

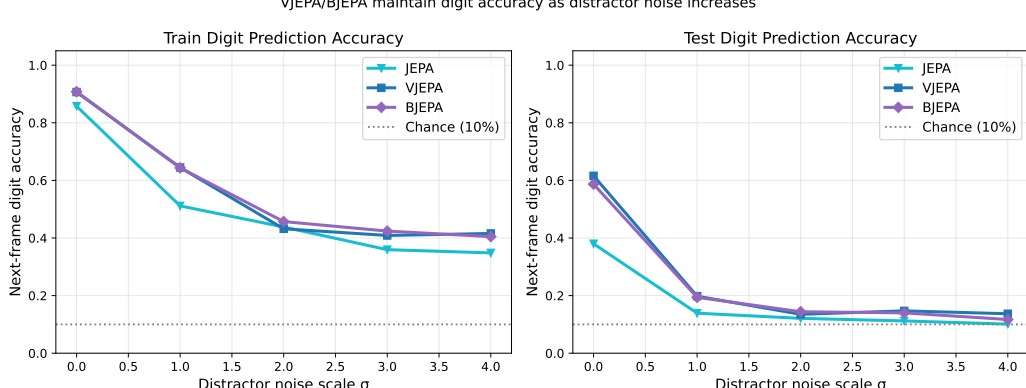

*Figure 9.* Temporal MNIST samples. **Row 1:** clean signal (digit class cycling $0 \rightarrow \cdots \rightarrow 9$). **Row 2:** random MNIST distractor images. **Row 3:** combined observation at $\sigma{=}1$: the target digit class remains identifiable but the background is corrupted.

*Figure 10.* Temporal MNIST: next digit-class prediction accuracy (train left, test right) across distractor scales. VJEPA and BJEPA consistently outperform JEPA, particularly at $\sigma{=}0$ where the probabilistic objective yields substantially richer representations (61.6% vs. 38.0%).

## K.5. CNN Stability and VICReg Regularization

In the linear and MLP experiments, VJEPA can be trained directly with the latent NLL plus target-side KL objective. In high-capacity CNN settings, however, we observed two practical instabilities: a trivial high-uncertainty solution, where the predictive variance grows rather than learning a useful mean, and unstable/collapsed target representations. Therefore, in the CNN experiments all JEPA-family models use VICReg-style variance/covariance stabilization in addition to their predictive objectives. This keeps the comparison focused on the predictive-vs-reconstructive distinction rather than on avoidable CNN collapse pathologies.

## K.6. Original Single-Seed Noisy-TV Results

The original Noisy-TV run uses seed 111, 6 000 training epochs, and all nine distractor scales. It is included for reproducibility and to show the full scale-by-scale trajectory. The main paper reports the more reliable $N{=}5$ multi-seed replication over three representative scales, while Table 11 preserves the complete original single-seed table.

*Table 10.* BJEPA zero-shot goal conditioning on Temporal MNIST. *No goal* = static prior (Exp. 3 baseline, 56.2%). *Delta* = Correct goal − No goal. Wrong goal at tight $\sigma^2_{\text{goal}}$ falls far below the no-goal baseline, showing the PoE mechanism amplifies goal signal regardless of correctness.

| $\sigma^2_{\text{goal}}$ | No goal | Correct goal | Wrong goal | Delta |
|---|---|---|---|---|
| 0.10 | 56.2% | **100.0%** | 15.6% | +43.8% |
| 0.30 | 56.2% | **100.0%** | 8.3% | +43.8% |
| 1.00 | 56.2% | 91.7% | 27.9% | +35.5% |
| 3.00 | 56.2% | 72.5% | 50.5% | +16.3% |
| 10.00 | 56.2% | 61.8% | 54.8% | + 5.6% |

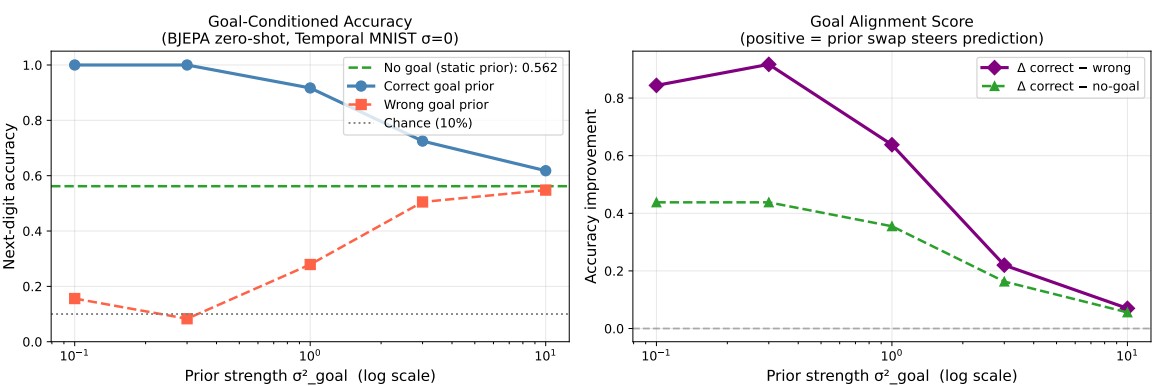

*Figure 11.* BJEPA zero-shot goal conditioning. **Left:** classification accuracy vs. goal prior variance $\sigma^2_{\text{goal}}$ for the three conditions (no goal, correct goal, wrong goal). **Right:** delta (correct − no goal) as a function of $\sigma^2_{\text{goal}}$. The PoE mechanism modulates goal influence monotonically: tight priors ($\sigma^2_{\text{goal}} \leq 0.3$) achieve perfect accuracy under a correct goal and strongly degrade under a wrong goal; diffuse priors ($\sigma^2_{\text{goal}} \to 10$) converge back to the no-goal baseline.

### K.7. Experiment 5: Moving MNIST CNN Benchmark

Moving MNIST tests the same idea in a nonlinear image-sequence setting with CNN encoders. A $32{\times}32$ canvas contains a single bouncing MNIST digit resized to $14{\times}14$. The latent state contains horizontal/vertical position, horizontal/vertical velocity, and digit class. Pixel noise $\sigma \in \{0, 0.5, 1.0\}$ is added to each frame. Five models are evaluated: JEPA-CNN, VJEPA-CNN, BJEPA-CNN, $\beta$-VAE-CNN, and RSSM-CNN, all using a lightweight CNN encoder with $D_z{=}16$.

Table 12 reports linear-probe class, position, and velocity recovery, and Figure 12 visualizes the same metrics across noise levels. The key observation is that BJEPA is the most stable JEPA-family model under CNN noise, while velocity remains near unrecoverable from a single frame for all models, as expected.

### K.8. Experiment 6: RSSM Baseline on Noisy TV

This experiment adds a Dreamer-style RSSM baseline to the linear Noisy-TV environment to test whether an explicit recurrent latent world model with observation reconstruction can overcome nuisance distractors. The RSSM is a minimal VRNN-style model with a GRU deterministic state, Gaussian posterior/prior over $z_t$, and Gaussian decoder $p(x_t \mid z_t, h_t)$. Dimensions match the Noisy-TV setup ($D_x{=}20$, $D_z{=}4$), and the model is trained by an ELBO with observation reconstruction and KL regularization.

Table 13 and Figure 13 show that the RSSM fits nuisance-specific observation structure but generalizes poorly on held-out signal recovery: signal $R^2$ is negative on the test trajectory across scales, while JEPA/VJEPA preserve substantially more predictive signal information.

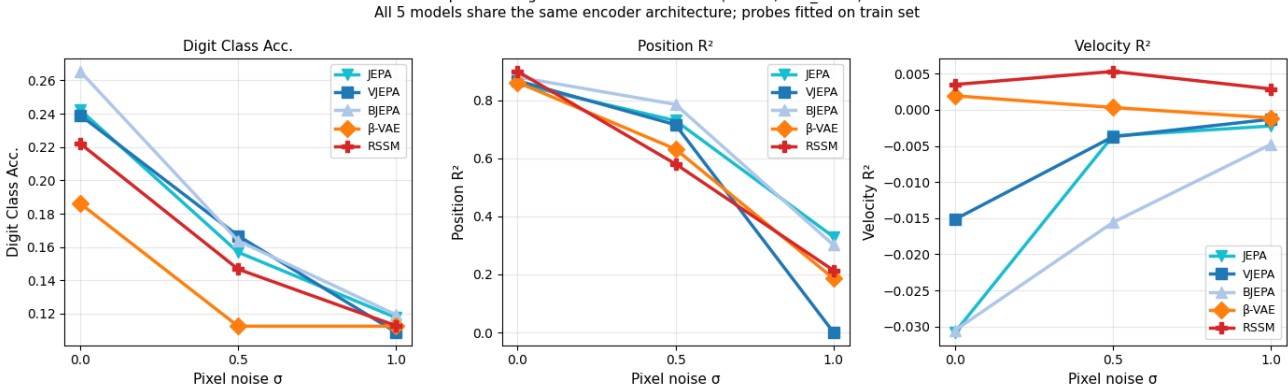

*Figure 12.* **Experiment 5: Moving MNIST CNN Benchmark.** Linear-probe performance (digit class accuracy, position $R^2$, velocity $R^2$) across three pixel noise levels $\sigma \in \{0, 0.5, 1.0\}$ for five models. BJEPA (light blue triangle) is the most noise-robust JEPA-family model. VJEPA (dark blue square) collapses at $\sigma{=}1.0$ (position $R^2 \to 0.001$). Velocity $R^2 \approx 0$ for all models: velocity is unrecoverable from a single frame.

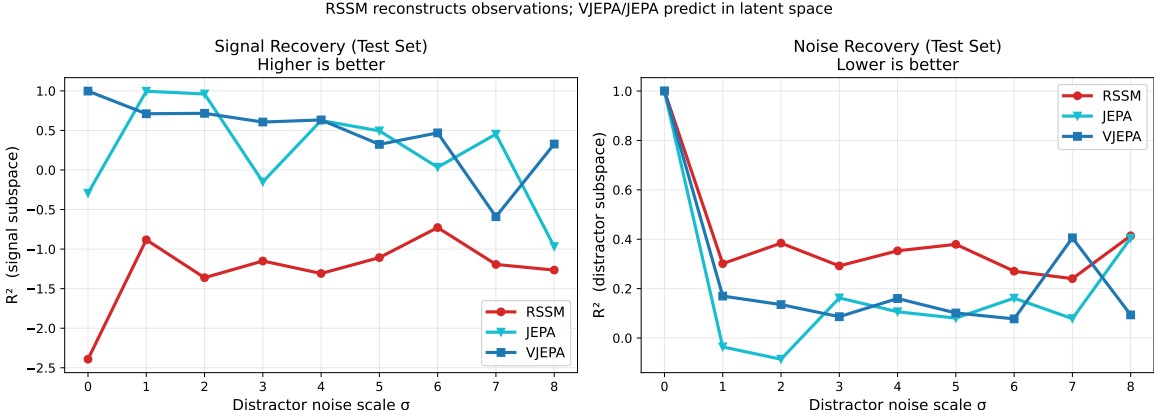

*Figure 13.* **Experiment 6: RSSM vs. JEPA vs. VJEPA on Noisy TV.** Signal $R^2$ (left, higher is better) and Noise $R^2$ (right, lower is better) on the held-out test trajectory across nine distractor scales. RSSM (red circle) achieves positive Signal $R^2$ on the training trajectory but *negative* Signal $R^2$ on the test trajectory at every scale, indicating trajectory-specific overfitting. JEPA (cyan triangle) and VJEPA (blue square) generalise substantially better.

## K.9. Experiment 7: ViT Encoder on STL-10

This experiment tests whether the predictive-objective advantage persists with a transformer image encoder and natural images. We train ViT-Tiny encoders from scratch on STL-10, resizing images to $64{\times}64$ and generating temporal class-cycling sequences analogous to Temporal MNIST. At each step, a randomly sampled distractor image is mixed with the signal image at scale $\sigma \in \{0, 0.5, 1.0, 2.0\}$. Evaluation again uses a logistic probe on predicted latents for next-class prediction.

Table 14 reports test accuracy, Figure 14 plots train/test accuracy across noise levels, and Figure 15 shows VJEPA's per-dimension predictive variance. Absolute accuracies are modest because the ViT is trained from scratch on a small labelled subset, but BJEPA/VJEPA are stronger than JEPA at moderate-to-high distractor scales, suggesting that the effect is not restricted to MLP or CNN backbones.

## K.10. Experiment 8: VJEPA-MPC on DM Control Suite

This experiment evaluates whether latent predictive models can support model-predictive control in a continuous-action environment. We use DMC Cheetah-run with $64{\times}64{\times}3$ pixel observations and Gaussian visual background noise $\sigma \in$

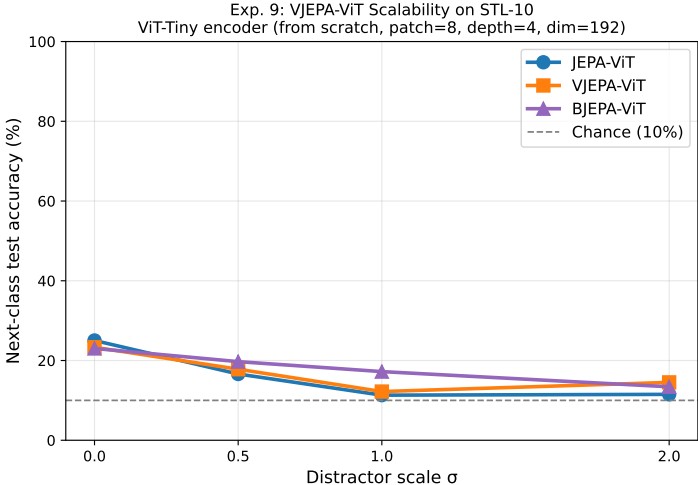

*Figure 14.* Experiment 7: ViT encoder on STL-10 - next object-class prediction accuracy (train left, test right) across distractor scales $\sigma$. BJEPA-ViT dominates at $\sigma \geq 0.5$; VJEPA-ViT outperforms JEPA-ViT at $\sigma \geq 1.0$. The BJEPA$\geq$VJEPA$>$JEPA ordering mirrors the Temporal MNIST result (Table 9), suggesting that the nuisance-robustness pattern is not restricted to MLP/CNN backbones.

$\{0, 0.5, 1.0\}$. VJEPA-MPC and JEPA-MPC use the same CNN encoder and action-conditioned latent predictor; Dreamer-lite uses a comparable latent transition model plus pixel decoder trained by an ELBO. Planning uses CEM over latent rollouts.

Table 15 lists the control hyperparameters, Table 16 reports final mean episode returns, and Figures 16 and 17 summarize learning and robustness. VJEPA-MPC achieves the best return at $\sigma=0$ and remains slightly above JEPA-MPC at $\sigma=0.5$, while Dreamer-lite stays near the random-policy baseline in this implementation.

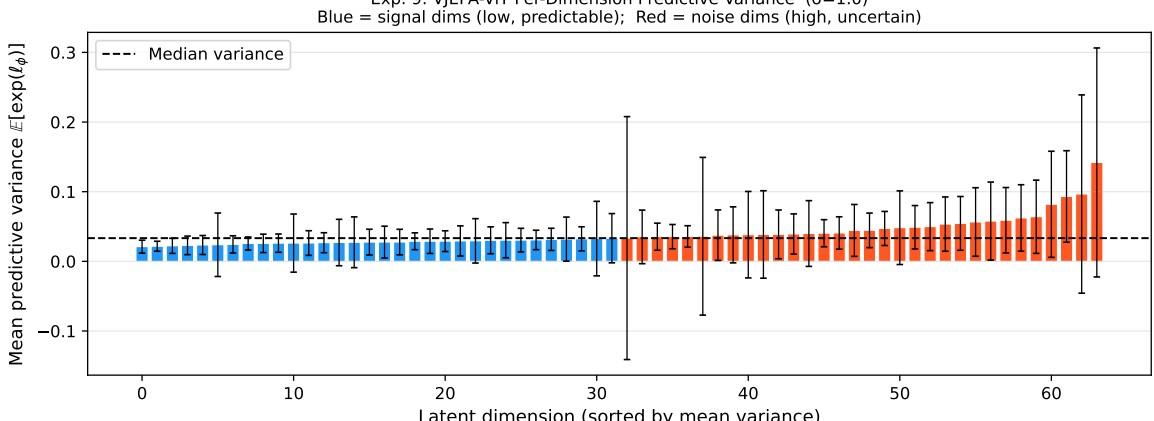

*Figure 15.* VJEPA-ViT per-dimension predictive variance at $\sigma \in \{0.0, 1.0, 2.0\}$. As distractor intensity grows, the model partitions its 64 latent dimensions: signal dimensions maintain low variance while noise dimensions converge towards the prior, reproducing the dimension-wise separation observed in Experiment 1 (Figure 5) for a ViT-encoded representation of natural images.

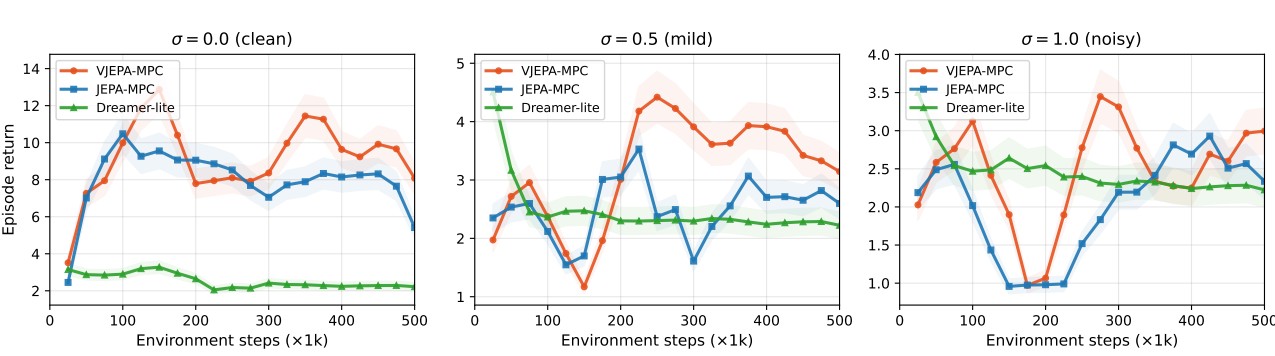

*Figure 16.* **Experiment 8: learning curves** on DMC Cheetah-run. Each panel shows a different noise level $\sigma$. VJEPA-MPC and JEPA-MPC both learn above the random-policy baseline at $\sigma=0$; Dreamer-lite remains near random throughout all conditions. Curves are 3-point moving averages for readability.

| Scale (SNR) | Model | ↑Signal $R^2$ (Tr/Te) | ↓Noise $R^2$ (Tr/Te) | ↓Time (Tr/Te) |
|---|---|---|---|---|
| 0.0 (inf dB) | VAE | **1.000** / **1.000** | NA / NA | 12.6s / 0.01s |
| 0.0 (inf dB) | AR | 0.999 / 0.999 | NA / NA | **6.4s** / 0.01s |
| 0.0 (inf dB) | JEPA | 0.947 / 0.930 | NA / NA | 16.9s / 0.01s |
| 0.0 (inf dB) | VJEPA | 0.999 / 0.999 | NA / NA | 13.9s / 0.01s |
| 0.0 (inf dB) | BJEPA | 0.987 / 0.981 | NA / NA | 23.5s / 0.01s |
| 1.0 (15.8 dB) | VAE | 0.978 / 0.973 | 0.065 / 0.002 | 12.6s / 0.01s |
| 1.0 (15.8 dB) | AR | 0.983 / 0.980 | 0.041 / -0.007 | **6.6s** / 0.01s |
| 1.0 (15.8 dB) | JEPA | 0.947 / 0.930 | 0.230 / 0.178 | 16.9s / 0.01s |
| 1.0 (15.8 dB) | VJEPA | **0.999** / **0.998** | **0.006** / **-0.012** | 13.8s / 0.01s |
| 1.0 (15.8 dB) | BJEPA | 0.975 / 0.976 | 0.029 / -0.003 | 23.5s / 0.01s |
| 2.0 (9.8 dB) | VAE | 0.903 / 0.853 | 0.283 / 0.194 | 12.8s / 0.01s |
| 2.0 (9.8 dB) | AR | 0.923 / 0.889 | 0.179 / 0.094 | **6.4s** / 0.01s |
| 2.0 (9.8 dB) | JEPA | 0.947 / 0.930 | 0.213 / 0.172 | 17.0s / 0.01s |
| 2.0 (9.8 dB) | VJEPA | **0.996** / **0.995** | **0.020** / -0.012 | 14.2s / 0.01s |
| 2.0 (9.8 dB) | BJEPA | 0.966 / 0.967 | 0.042 / **-0.023** | 23.3s / 0.01s |
| 3.0 (6.3 dB) | VAE | 0.852 / 0.770 | 0.458 / 0.393 | 12.5s / 0.01s |
| 3.0 (6.3 dB) | AR | 0.870 / 0.801 | 0.328 / 0.256 | **6.5s** / 0.01s |
| 3.0 (6.3 dB) | JEPA | 0.916 / 0.841 | **0.000** / **-0.003** | 16.8s / 0.01s |
| 3.0 (6.3 dB) | VJEPA | **0.947** / **0.930** | 0.224 / 0.176 | 13.5s / 0.01s |
| 3.0 (6.3 dB) | BJEPA | 0.946 / 0.919 | 0.106 / 0.078 | 23.0s / 0.01s |
| 4.0 (3.8 dB) | VAE | 0.822 / 0.730 | 0.512 / 0.458 | 12.5s / 0.01s |
| 4.0 (3.8 dB) | AR | 0.839 / 0.756 | 0.394 / 0.338 | **6.5s** / 0.01s |
| 4.0 (3.8 dB) | JEPA | **0.999** / **0.999** | **0.004** / **-0.010** | 16.6s / 0.02s |
| 4.0 (3.8 dB) | VJEPA | 0.994 / 0.993 | 0.025 / -0.007 | 13.8s / 0.01s |
| 4.0 (3.8 dB) | BJEPA | 0.920 / 0.899 | 0.213 / 0.156 | 23.2s / 0.01s |
| 5.0 (1.8 dB) | VAE | 0.777 / 0.681 | 0.543 / 0.492 | 12.5s / 0.01s |
| 5.0 (1.8 dB) | AR | 0.806 / 0.717 | 0.418 / 0.364 | **6.4s** / 0.01s |
| 5.0 (1.8 dB) | JEPA | **0.947** / **0.930** | **0.200** / **0.183** | 16.5s / 0.01s |
| 5.0 (1.8 dB) | VJEPA | 0.945 / 0.928 | 0.227 / 0.190 | 14.1s / 0.01s |
| 5.0 (1.8 dB) | BJEPA | 0.884 / 0.841 | 0.317 / 0.240 | 22.3s / 0.01s |
| 6.0 (0.3 dB) | VAE | 0.729 / 0.627 | 0.577 / 0.531 | 12.3s / 0.01s |
| 6.0 (0.3 dB) | AR | 0.770 / 0.677 | 0.437 / 0.386 | **7.1s** / 0.01s |
| 6.0 (0.3 dB) | JEPA | **0.999** / **0.999** | **0.007** / **-0.011** | 16.0s / 0.01s |
| 6.0 (0.3 dB) | VJEPA | 0.947 / 0.930 | 0.197 / 0.202 | 13.7s / 0.01s |
| 6.0 (0.3 dB) | BJEPA | 0.935 / 0.914 | 0.205 / 0.171 | 22.8s / 0.01s |
| 7.0 (-1.1 dB) | VAE | 0.675 / 0.566 | 0.615 / 0.574 | 12.7s / 0.01s |
| 7.0 (-1.1 dB) | AR | 0.729 / 0.631 | 0.462 / 0.416 | **7.1s** / 0.01s |
| 7.0 (-1.1 dB) | JEPA | **0.946** / **0.929** | **0.209** / 0.191 | 16.1s / 0.01s |
| 7.0 (-1.1 dB) | VJEPA | 0.939 / 0.918 | 0.264 / **0.182** | 13.4s / 0.01s |
| 7.0 (-1.1 dB) | BJEPA | 0.894 / 0.851 | 0.291 / 0.237 | 23.1s / 0.01s |
| 8.0 (-2.2 dB) | VAE | 0.613 / 0.499 | 0.656 / 0.620 | 12.3s / 0.01s |
| 8.0 (-2.2 dB) | AR | 0.680 / 0.578 | 0.491 / 0.449 | **7.1s** / 0.01s |
| 8.0 (-2.2 dB) | JEPA | **0.947** / **0.930** | **0.226** / **0.183** | 16.1s / 0.01s |
| 8.0 (-2.2 dB) | VJEPA | 0.905 / 0.870 | 0.299 / 0.251 | 13.4s / 0.01s |
| 8.0 (-2.2 dB) | BJEPA | 0.896 / 0.841 | 0.292 / 0.238 | 23.0s / 0.01s |

*Table 11.* **Original single-seed run** (seed = 111, 6 000 epochs) across all nine noise scales. Included for reproducibility; see Table 3 for the multi-seed ($N$=5) further results. $Tr/Te$: training/test set. Underlined bold = best per scale.

| Model | $\sigma$ | $\uparrow$ Class Acc | $\uparrow$ Pos $R^2$ | Vel $R^2$ | Time (s) |
|---|---|---|---|---|---|
| JEPA-CNN | | 0.242 | 0.856 | $-0.031$ | 1 426 |
| VJEPA-CNN | | 0.239 | 0.866 | $-0.015$ | 1 810 |
| BJEPA-CNN | 0.0 | **0.265** | **0.879** | $-0.031$ | 2 306 |
| $\beta$-VAE-CNN | | 0.186 | 0.861 | 0.002 | **73** |
| RSSM-CNN | | 0.222 | 0.899 | 0.004 | 3 853 |
| JEPA-CNN | | 0.157 | 0.729 | $-0.004$ | 764 |
| VJEPA-CNN | | 0.167 | 0.715 | $-0.004$ | 1 016 |
| BJEPA-CNN | 0.5 | **0.164** | **0.785** | $-0.016$ | **999** |
| $\beta$-VAE-CNN | | 0.113 | 0.631 | 0.000 | **39** |
| RSSM-CNN | | 0.147 | 0.580 | 0.005 | 2 108 |
| JEPA-CNN | | 0.118 | 0.330 | $-0.002$ | 591 |
| VJEPA-CNN | | 0.109 | 0.001 | $-0.001$ | 693 |
| BJEPA-CNN | 1.0 | **0.119** | **0.302** | $-0.005$ | **586** |
| $\beta$-VAE-CNN | | 0.113 | 0.186 | $-0.001$ | **30** |
| RSSM-CNN | | 0.113 | 0.214 | 0.003 | 2 121 |

*Table 12.* **Experiment 5: Moving MNIST CNN Benchmark.** Linear-probe results across three noise levels, seed $= 111$, $D_z = 16$. **Bold** = best among JEPA-family per noise level; bold = overall best. Vel $R^2 \approx 0$ for all models: velocity cannot be determined from a single frame.

| Scale (SNR) | RSSM | | JEPA | | VJEPA | |
|---|---|---|---|---|---|---|
| | Sig $R^2$ | Noi $R^2$ | Sig $R^2$ | Noi $R^2$ | Sig $R^2$ | Noi $R^2$ |
| 0.0 (92 dB) | $-2.392$ | 1.000 | $-0.294$ | 1.000 | **0.998** | 1.000 |
| 1.0 (15.8 dB) | $-0.882$ | 0.301 | **0.995** | $-0.036$ | 0.711 | 0.170 |
| 2.0 (9.8 dB) | $-1.362$ | 0.384 | **0.961** | $-0.086$ | 0.716 | 0.135 |
| 3.0 (6.3 dB) | $-1.150$ | 0.292 | $-0.150$ | 0.162 | **0.606** | 0.086 |
| 4.0 (3.8 dB) | $-1.308$ | 0.353 | 0.625 | 0.106 | **0.633** | 0.160 |
| 5.0 (1.8 dB) | $-1.108$ | 0.379 | **0.494** | 0.081 | 0.324 | 0.101 |
| 6.0 (0.3 dB) | $-0.728$ | 0.271 | 0.035 | 0.161 | **0.469** | 0.077 |
| 7.0 ($-1.1$ dB) | $-1.194$ | 0.240 | 0.453 | 0.078 | $-0.590$ | 0.406 |
| 8.0 ($-2.2$ dB) | $-1.265$ | 0.414 | $-0.968$ | 0.405 | **0.328** | 0.094 |

*Table 13.* **Experiment 6: RSSM vs. JEPA vs. VJEPA on Noisy TV** (seed $= 111$, 3 000 epochs, test set). Signal $R^2$ (higher is better) and Noise $R^2$ (lower is better). **Bold** = highest Signal $R^2$ per scale. Note: RSSM probes the current-step signal $s_t$; JEPA/VJEPA probe the next-step signal $s_{t+1}$, consistent with their respective objectives. Training times: RSSM 4 200–4 700 s per scale; JEPA/VJEPA 7–8 s per scale.

*Table 14.* **Experiment 7: ViT encoder on STL-10.** Next object-class test accuracy (%) from a logistic probe on predicted latents. Chance = 10%. Bold = best per noise scale. Results confirm architecture-agnostic nuisance invariance.

| Model | $\sigma{=}0.0$ | $\sigma{=}0.5$ | $\sigma{=}1.0$ | $\sigma{=}2.0$ |
|---|---|---|---|---|
| JEPA-ViT | **25.0** | 16.6 | 11.3 | 11.5 |
| VJEPA-ViT | 23.3 | 17.8 | 12.2 | **14.5** |
| BJEPA-ViT | 23.0 | **19.7** | **17.2** | 13.4 |
| Chance | 10.0 | 10.0 | 10.0 | 10.0 |

*Table 15.* Hyperparameters for Experiment 8 (DM Control MBRL).

| Parameter | Value |
| --- | --- |
| Latent dimension $Z$ | 256 |
| Predictor hidden | 512 |
| Batch size | 512 |
| Replay buffer | 100 000 |
| World model LR | $10^{-4}$ (Adam) |
| Reward model LR | $10^{-3}$ (Adam) |
| EMA $\tau$ | 0.995 |
| CEM horizon $H$ | 10 |
| CEM samples $N$ | 1024 |
| CEM elites | 100 (10%) |
| CEM iterations $K$ | 3 |
| Random exploration | 20 000 steps |
| Total steps | 500 000 |
| Evaluation | every 25 000 steps, 5 episodes |

*Table 16.* Experiment 8: mean episode return (avg. last 5 evals, steps 400k–500k) on DMC Cheetah-run. Random-policy baseline $\approx$2.3. $\Delta$: relative drop from $\sigma$=0.

| Model | $\sigma$=0.0 | $\sigma$=0.5 | $\Delta_{0.5}$ | $\sigma$=1.0 | $\Delta_{1.0}$ |
| --- | --- | --- | --- | --- | --- |
| VJEPA-MPC | **9.5** | **3.6** | $-63\%$ | **2.7** | $-71\%$ |
| JEPA-MPC | 7.8 | 2.8 | $-64\%$ | 2.6 | $-66\%$ |
| Dreamer-lite | 2.3 | 2.3 | $0\%$ | 2.3 | $0\%$ |
| Random policy | | | $\approx 2.3$ | | |

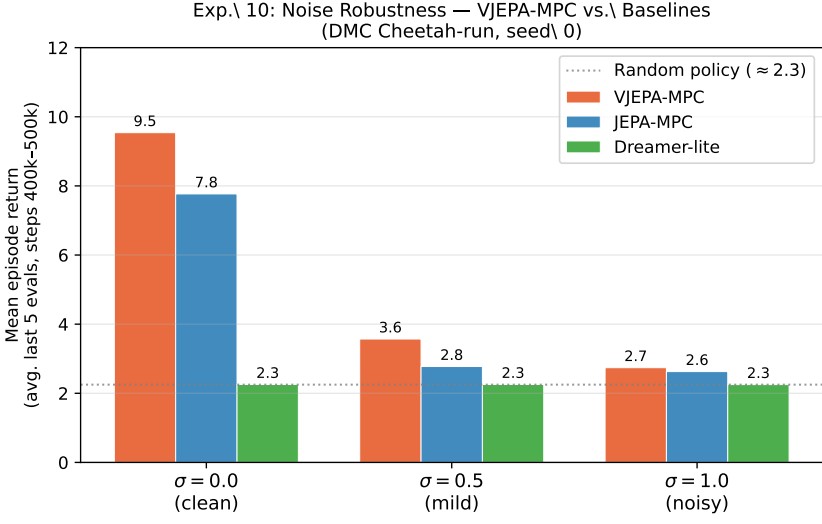

*Figure 17.* **Experiment 8: noise-robustness summary.** Mean episode return (last 5 evals) at each $\sigma$ level. VJEPA-MPC achieves the highest return at $\sigma$=0 (9.5) and maintains a slight advantage over JEPA-MPC at $\sigma$=0.5 (3.6 vs. 2.8). Both predictive models degrade under noise. Dreamer-lite matches the random-policy baseline ($\approx$2.3) at *every* noise level, consistent with the difficulty of learning reward-predictive latents from pixel-ELBO training in this lightweight setup. Dashed line: random-policy baseline.

