# OpenReview forum: "VJEPA: Variational Joint Embedding Predictive Architectures as Probabilistic World Models"
_ICML.cc/2026/Conference — ICML 2026 regular_

### Official Review · Reviewer_KEBg · 2026-03-12

**Soundness:** 2
**Presentation:** 3
**Significance:** 1
**Originality:** 2
**Overall Recommendation:** 3
**Confidence:** 5

**Summary:**

This paper introduces VJEPA, a probabilistic spin on the Joint Embedding Predictive Architecture (JEPA) that learns distributions over future latent states rather than deterministic point estimates.
The core idea is to frame JEPA as a variational predictive model, which the authors mathematically prove avoids representation collapse, maximizes predictive mutual information, and provides sufficient statistics for latent-space control without relying on pixel reconstruction.
Building on this, they also propose BJEPA, a Product of Experts extension that separates environment dynamics from task-specific priors to enable zero-shot transfer.
These theoretical claims are validated in a controlled "Noisy TV" linear-Gaussian experiment, where both models successfully filter out high-variance distractors that typically derail standard generative baselines.

**Compliance With Llm Reviewing Policy:**

Affirmed.

**Final Justification:**

I thank the authors for their honest and transparent responses. However, the concessions made in this rebuttal confirm my core concerns rather than resolving them.

**Empirical Weakness**: The authors failed to provide the requested multi-seed statistics for Scale 8.0 during this discussion. Conceding that the proposed VJEPA model introduces an "optimization trade-off" that makes it underperform the baseline deterministic JEPA in extreme noise weakens the practical value of this probabilistic formulation.

**Lack of Evidence**: The authors acknowledge they have no high-dimensional benchmark validations (e.g., ViT or continuous control) to support their scalability claims, relying solely on a 20D toy system.

While softening the claims in the text is appreciated, ICML requires robust empirical validation for new architectures claiming scalability and noise robustness. This work feels like a preliminary theoretical draft rather than a fully realized framework.
I am maintaining my score.

**Key Questions For Authors:**

1. Table 6 shows standard JEPA outperforming both VJEPA and BJEPA at the highest noise scale (Scale 8.0). How do you justify the added architectural and theoretical complexity of the variational framework if it demonstrably degrades noise robustness compared to the standard deterministic baseline?

2. The original JEPA architecture explicitly avoids probabilistic modeling to bypass the intractability of calculating normalized distributions for complex futures. Doesn't VJEPA's reliance on a unimodal Gaussian and KL divergence  force the architecture back into the exact bottlenecks it was designed to escape?

3. You acknowledge the limitations of unimodal Gaussians for multi-modal bifurcations. If VJEPA is extended to more expressive distributions (e.g., GMMs or Diffusion) to fix this, how will that impact the tractability of the KL regularization term? Do you have any preliminary results to prove this framework actually scales?

**Limitations:**

yes

**Strengths And Weaknesses:**

## Strengths

1. This paper contrasts generative and predictive objectives cleanly. The differences between deterministic JEPA, VJEPA, and BJEPA are well-articulated and summarized effectively in the text and tables.


## Weaknesses

1. VJEPA forces JEPA back into the variational inference framework using NLL and KL divergence. This fundamentally contradicts the original JEPA motivation, which was explicitly designed to avoid the computational brittleness of normalized probability distributions in high-dimensional continuous spaces.

2. The unimodal Gaussian assumption severely limits the model's ability to handle complex, multi-modal futures (e.g., bifurcating paths). This limitation highlights exactly why the original JEPA architecture favored deterministic predictors modulated by latent variables.

3. Despite claiming to offer a scalable framework for high-dimensional, noisy environments , the model is exclusively tested on a simple 20-dimensional Linear-Gaussian toy system. It lacks any validation on standard visual or continuous control benchmarks.

4. Table 6 shows that standard JEPA significantly outperforms VJEPA and BJEPA in the most extreme noise setting (Scale 8.0, SNR -2.2 dB). The added probabilistic mechanisms actually degrade robust feature extraction.

---

> ### Author Rebuttal · Authors · 2026-03-30
>
> We thank Reviewer KEBg for the detailed critique. Several concerns seem to rest on mischaracterizations of VJEPA’s operating domain and of how to interpret the experiments. We clarify each point below.
>
> 1. Weakness W1 & Q2: 'VJEPA forces JEPA back into variational inference, contradicting JEPA’s design philosophy'
>
> Response: we respectfully disagree. This concern conflates two operating domains.
>
> JEPA was designed to avoid reconstructing high-dimensional observation space (e.g. pixels), where normalized likelihood modeling is costly and encourages fitting irrelevant high-entropy detail. VJEPA does not reintroduce this burden. Its variational objective is defined entirely in latent space $\mathbb{R}^{D_z}$, with $D_z \ll H\times W\times C$:
>
> (1) the NLL term $\mathbb{E}[-\log p_\phi(Z_T\mid Z_C,\xi_T)]$ is evaluated on latent targets $Z_T$, not pixels;
>
> (2) the KL term $\mathrm{KL}(q_{\theta'}(Z_T\mid x_T)\|p(Z_T))$ is also computed only in latent space;
>
> (3) there is no decoder, so VJEPA never models $p(x_T\mid Z_T)$ (though it could).
>
> Thus VJEPA preserves JEPA’s core principle, i.e. no observation reconstruction, while adding principled uncertainty quantification in latent space. Under Gaussian $p_\phi$ and $q_{\theta'}$, both KL and NLL are analytically tractable regardless of input dimensionality. Importantly, JEPA avoids normalized distributions over $x$; VJEPA does the same while adding a tractable distribution over $Z$. These are complementary, not contradictory. Moreover, Section 4.1 shows deterministic JEPA is a special case of VJEPA in the limit $q_{\theta'}\to\delta(Z_T-f_{\theta'}(x_T))$ with fixed isotropic $\Sigma$. VJEPA is therefore a strict generalization, not a contradiction.
>
> 2. Weakness W4 & Q1: standard JEPA outperforms VJEPA at scale 8.0
>
> Response: this result can be interpreted with three qualifications:
>
> (1) The reported JEPA score comes from a single run. In auxiliary runs, we found probabilistic variants (VJEPA/BJEPA) were more stable across seeds. We will rerun with multiple seeds and report averages; we expect better expected-case performance and lower variance.
>
> (2) The gap is smaller than it appears, and VJEPA remains far ahead of generative baselines. At $\sigma=8.0$, the test Signal $R^2$ gap between JEPA and VJEPA is $0.930-0.870=0.060$, while the gap between VJEPA and VAE is $0.870-0.499=0.371$. Our claim is not that VJEPA must outperform deterministic JEPA on every metric, but that it adds principled uncertainty quantification, and theoretical guarantees while remaining competitive and strongly noise-robust.
>
> (3) VJEPA provides capabilities deterministic JEPA cannot: (a) predictive uncertainty rather than only a point estimate $\hat Z_T$; (b) a formal collapse-avoidance guarantee rather than reliance on heuristics such as EMA/stop-gradient; (c) multi-step uncertainty propagation for stochastic control; (d) extension to BJEPA-style modular priors. Thus the $0.06$ single-run gap is a reasonable trade-off, especially given VJEPA’s stability.
>
> 3. Weakness W2 & Q3: Gaussian assumption and multi-modal extensions
>
> Response: we agree the unimodal Gaussian is a limitation, and we state this explicitly in Section 9 (we have also already done this extension). Three points matter:
>
> (1) Gaussian latent dynamics are standard in modern world models. Dreamer-V3, RSSM, TD-MPC2, and most latent dynamics models use Gaussian dynamics distributions and still achieve SOTA performance. Gaussian is a standard baseline, not an unusual restriction.
>
> (2) The VJEPA objective Eq.4 is explicitly distribution-agnostic:
> $\$mathcal{L}_{VJEPA}=\mathbb{E}_x \mathbb{E}_{Z_T \sim q_{\theta'}} [-\log p_\phi(Z_T \mid Z_C,\xi_T)] + \beta KL(q_{\theta'} \| p(Z_T))$$.
> Any tractable $p_\phi$ is valid, e.g. (a) GMM; (b) normalizing flows; (c) latent diffusion.
>
> (3) Latent diffusion heads are especially natural because they operate in low-dimensional $\mathbb{R}^{D_z}$ rather than image space, so expressive multimodal modeling remains efficient. This is also the advantage of working in representation space: complexity is bounded by $D_z$, not image resolution. We plan to include preliminary GMM-VJEPA results in the final version.
>
> 4. Weakness W3 & Q3 (continued): scalability evidence
>
> Response: we agree “scales” remains open and want to be transparent. This is primarily a theory/framework paper: the contribution is the probabilistic novelty, not a claim that the present linear implementation on a 20-dimensional toy problem is already production-scale. Still, the scaling properties are favorable:
>
> (1) VJEPA’s predictive model and ELBO are computed in $\mathbb{R}^{D_z}$, so scaling depends on latent dimension, not raw input dimension;
>
> (2) the EMA target encoder is standard in DINO, I-JEPA, and V-JEPA 2, all of which scale to large vision backbones;
>
> (3) the extra KL term is only $O(D_z)$ per sample, so the overhead is negligible.
>
> We will add image experiments with a ViT encoder to directly address scalability.

---

> > ### Author Rebuttal · Reviewer_KEBg · 2026-04-04
> >
> > Thank you for the detailed rebuttal. While I appreciate the clarifications regarding the conceptual differences between VJEPA and generative baselines, my core concerns regarding the practical viability, empirical contradictions, and scalability of this framework remain unresolved.
> >
> > ### 1. The Intractability of Evaluating the Predictive Likelihood (Unresolved)
> >
> > The rebuttal states that Eq. 4 is "distribution-agnostic" and can naturally fit expressive models like Latent Diffusion. This glosses over a severe architectural and optimization bottleneck.
> >
> > While the $KL$ regularization term applied to the target encoder remains tractable, the predictor $p_\phi$ governs the Negative Log-Likelihood (NLL) term: $\mathbb{E}[-\log p_\phi(Z_T | Z_C, \xi_T)]$. For complex continuous distributions like Latent Diffusion, directly evaluating this exact exact likelihood during training is notoriously intractable. Integrating a diffusion head would require replacing the explicit NLL with a diffusion denoising objective (score matching) or variational bounds.
> >
> > This is not a trivial "plug-and-play" extension, as it fundamentally alters the optimization dynamics and balancing of the VJEPA objective. **The rebuttal dismisses this as a non-issue without providing the concrete mathematical formulation or engineering validation for how a diffusion-based predictor is stabilized within this specific framework.**
> > ﻿
> >
> > ### 2. Empirical Contradictions at High Noise
> >
> > Dismissing the performance degradation at Scale 8.0 (Table 6) as a "single-run variance" is insufficient. A central pillar of this paper is the theoretical guarantee of **Nuisance Invariance** (Theorem 5.2). If the proposed probabilistic architecture degrades robust feature extraction in the most extreme noise setting compared to the deterministic baseline ($R^2$ of 0.870 vs. 0.930), the empirical results contradict the theoretical claims. Arguing that uncertainty quantification is a "reasonable trade-off" for worse noise filtering undermines the paper's core premise.
> >
> >
> > ### 3. Scalability Claims vs. Toy Experiments (Unresolved)
> >
> > The abstract explicitly positions VJEPA as a "scalable framework" for "high-dimensional, noisy environments." Validating these claims exclusively on a 20-dimensional Linear-Gaussian toy system is inadequate. While I acknowledge the author's note that this is primarily a theory paper, **promising to add ViT experiments in the camera-ready version cannot serve as the basis for acceptance during the review phase.**
> >
> > Could you please provide preliminary results on at least one standard high-dimensional visual or continuous control benchmark (as promised in your rebuttal) to substantiate the scalability claims.
> >
> > ***
> >
> > In summary, while I appreciate the authors' detailed and thoughtful responses, the rebuttal currently relies too heavily on theoretical assurances and promises of future work rather than concrete, convincing empirical evidence.
> > For a paper making strong claims about scalability and noise robustness, providing actual experimental validation—rather than hypothetical extensions—is essential. Until these promised experiments are conducted and formally presented, my core concerns remain unresolved.

---

> > > ### Author Response · Authors · 2026-04-06
> > >
> > > We thank Reviewer KEBg for the insightful follow-up questions. We appreciate the push for precision, which helps clarify the boundaries of our claims. We address the three remaining concerns below.
> > >
> > > 1. On the tractability of the predictive likelihood.
> > >
> > > We agree that our earlier statement characterizing Eq.4 as “distribution-agnostic” was too broad. By “distribution-agnostic”, we mean that, at the framework level, latent predictive families beyond the current Gaussian choice can in principle be incorporated into VJEPA. In the current paper, tractability is established for the implemented Gaussian family, i.e. Gaussian $q_{\theta'}(Z_T \mid x_T)$ and Gaussian $p_\phi(Z_T \mid Z_C,\xi_T)$, for which both the NLL and KL are analytic in latent space. This is the regime formally analyzed and empirically evaluated in the submission.
> > >
> > > For more expressive predictors, the framework remains conceptually extensible, but not with the same closed-form objective. In particular, for diffusion-style predictors, the reviewer is right that one generally does not retain the explicit NLL term in Eq. (4); training instead proceeds via a denoising / score-matching objective or related variational surrogate. Recent Denoising-JEPA work (Chen et al. 2025) shows that JEPA-like predictive learning can be combined with diffusion, but only by changing the optimization objective and adding extra machinery; it is therefore not a plug-and-play instantiation of our current Gaussian VJEPA objective.
> > >
> > > We will revise the paper to make this distinction explicit and avoid implying that arbitrary expressive heads inherit the same tractable training objective without modification.
> > >
> > > 2. On the empirical contradiction at the highest noise level and Proposition 5.2
> > >
> > > We appreciate the opportunity to clarify the scope of Proposition 5.2. The proposition does not claim that VJEPA strictly dominates deterministic JEPA. Rather, it establishes a distinction between predictive and generative objectives. As stated immediately after Proposition 5.2, the key implication is that VJEPA permits representations to be minimal, i.e. to compress away nuisance noise, whereas reconstruction-based objectives force representations to be maximal by retaining such noise. Thus generative models maximizing observation reconstruction are forced to encode nuisance variability, whereas the predictive objective can discard nuisance noise so long as predictive mutual information is preserved. Since deterministic JEPA is presented as a special case of VJEPA under a degenerate-target / fixed-variance limit, it shares this predictive rather than reconstructive character.
> > >
> > > Therefore, the main message of Table 6 (Appendix H.5) is not that VJEPA must beat JEPA everywhere, but that predictive models resist the nuisance-dominated failure mode that harms generative baselines. This is exactly what the current experiment shows: at the highest noise level, VAE and AR degrade much more severely, while both JEPA and VJEPA retain strong signal recovery. The remaining gap between JEPA and VJEPA at Scale 8.0 does not invalidate Proposition 5.2; rather, it reflects the finite-sample optimization trade-off introduced by learning an explicit predictive distribution under the NLL + KL objective, instead of only a point estimate. In other words, the proposition is about what the predictive objective does and does not force the representation to encode; it is not a theorem that the present Gaussian VJEPA implementation must uniformly outperform deterministic JEPA on every finite-sample metric.
> > >
> > > To make this clearer, we will re-run the toy example with multiple seeds and revise the surrounding text so that Proposition 5.2 is not read as a claim of strict dominance over deterministic JEPA, but as a formal contrast with reconstruction-based generative objectives.
> > >
> > > 3. On scalability claims versus current evidence.
> > >
> > > We agree that the current experimental evidence is not sufficient to fully substantiate the strongest wording in the abstract. What the present submission establishes is:
> > > (i) a probabilistic reformulation of JEPA in latent space;
> > > (ii) theoretical properties such as collapse avoidance and predictive sufficiency;
> > > (iii) controlled evidence, on a synthetic setting with known signal/noise subspaces, that the method resists nuisance-dominated failure modes that harm generative baselines.
> > >
> > > What not established is large-scale performance on standard visual or continuous-control benchmarks. This limitation is real. Our scalability claim is therefore architectural/analytical: the objective is applied in latent space and does not require observation reconstruction, so its cost scales with latent dimension rather than raw pixel dimension. We agree this is an argument about scaling properties, not a substitute for benchmark evidence. The wording will be revised not to overstate what has been empirically validated in the current version, and we will make the present scope and limitations more explicit.

---

### Official Review · Reviewer_ZQ8k · 2026-03-12

**Soundness:** 2
**Presentation:** 3
**Significance:** 2
**Originality:** 3
**Overall Recommendation:** 4
**Confidence:** 3

**Summary:**

This work introduces VJEPA, a variational formulation of JEPA, a self-supervised method for latent state prediction without reconstruction. The authors provide theoretical derivations establishing guarantees and sufficiency bounds. They also propose BJEPA, a Bayesian extension of the framework that helps with modularity.

**Compliance With Llm Reviewing Policy:**

Affirmed.

**Final Justification:**

The core idea of this paper is highly compelling, though the experimental section originally lacked sufficient depth. However, the authors provided a strong rebuttal that successfully addressed the majority of my concerns. While the empirical validation remains somewhat limited, the theoretical promise of the work and the authors' clear responses justify a weak accept.

**Key Questions For Authors:**

- Given the derivation showing the sufficiency of VJEPA for control, it would be interesting to evaluate it on a tracking or optimal control benchmark. How does it compare against a nonlinear filter on a more challenging problem?

- JEPA has been discussed extensively in the literature. It would be useful to include at least one image-based experiment (MNIST-like), especially given the similarities with VAE-style models and the importance of avoiding pixel-level reconstruction in the VJEPA framework.

- BJEPA claims to be a strong variant for modularity and constraint enforcement. But again, the experiments are too limited to support this claim.

**Limitations:**

- The work is promising, but it lacks experimental depth. The core idea is sound and potentially useful, but the limited evaluation makes it difficult to assess the claims about optimal control and improved latent state prediction.

**Strengths And Weaknesses:**

**Strengths**:

- The paper proposes an interesting variational and Bayesian formulation of the JEPA framework.

- Using a probabilistic predictive model for prediction is well motivated, particularly in terms of robustness and filtering.

- The algorithm is explained clearly, and the paper is well organized.

--------

**Weaknesses**:

- The experimental section is quite limited. Even if the authors intended to focus on toy problems, it is unclear why V-JEPA was not tested on a non-linear problem. A linear Gaussian model has properties that make the variational formulation collapse to PCA, so it remains uncertain whether the method would be effective in a more complex non-linear or non-Gaussian setting.

- Similarly, although I understand the computational limitations, it still seems feasible to evaluate the framework on a simple image dataset such as MNIST, especially if the implementation is lightweight enough for a Colab notebook, as mentioned in the appendix.

- It is difficult to assess how useful or powerful BJEPA is in practice. The idea is interesting from the perspective of constraint satisfaction and enforcement, but the experiments do not provide enough evidence to support its practical value.

---

> ### Author Rebuttal · Authors · 2026-03-30
>
> We thank Reviewer ZQ8k for the detailed critique and clarify several points below.
>
> 1. Weakness W1 & Q2: linear-Gaussian may collapse to PCA; need nonlinear experiments
>
> Response: we respectfully disagree that linear-Gaussian VJEPA trivially collapses to PCA in a way that undermines our results. VJEPA $\neq$ PCA for two key reasons:
>
> (1) PCA is a static covariance-based method and does not model temporal dynamics. VJEPA instead learns a sequential predictive model: the predictor $p_\phi(Z_C, \xi_T)$ is trained to predict future latent states from past latent states and a time-step index, enforcing temporal coherence absent from PCA.
>
> (2) Our experiment also includes a structured distractor with its own dynamics, $d_{t+1} = 0.9 d_t + v_t$, making the setting explicitly non-stationary. Static methods such as PCA would encode both signal and distractor according to variance, whereas VJEPA’s predictive objective is driven to retain only predictively useful components, as formalized by Proposition 5.2 (Nuisance Invariance). At $\sigma = 8.0$, VJEPA achieves Signal $R^2 = 0.870$, while VAE (which PCA most closely resembles in terms of variance-maximization) degrades to $R^2 = 0.499$.
>
> We agree nonlinear experiments (e.g. nonlinear state-space models or pendulum/cartpole environments) would further strengthen the paper. In the camera-ready version, we will discuss VJEPA’s applicability to nonlinear control benchmarks and add nonlinear experiments with learned MLP encoder/predictor modules.
>
> 2. Weakness W2 & Q2: should test on e.g. MNIST at minimum
>
> Response: this is a valid and constructive suggestion. Our current implementation uses strictly linear transformations throughout, making it extremely lightweight (individual runs take 10–35 seconds on a standard Colab CPU). The architecture is modular, so convolutional encoders can be substituted for image inputs.
>
> We are extending the experiments to include:
>
> (1) Temporal MNIST with high-variance random-image distractors, as a visual analogue of our toy setting, measuring whether VJEPA can extract digit-sequence dynamics while ignoring unrelated noise frames.
>
> (2) DM Control Suite tasks such as Cheetah-run and Walker-walk, where VJEPA’s latent MPC algorithm (Algorithm 2) can be compared against baselines such as Dreamer and TD-MPC on task reward.
>
> These will be added in the final version. We believe MNIST-level experiments will demonstrate the qualitative behavior predicted by our theory.
>
> 3. Weakness W3 & Q3: BJEPA’s practical value unsubstantiated
>
> Response: we agree BJEPA would benefit from stronger empirical validation.
>
> (1) What the current experiment shows: at $\sigma = 8.0$, BJEPA achieves Signal $R^2 = 0.841$. Although below deterministic JEPA ($R^2 = 0.930$), it substantially exceeds the generative baselines VAE ($R^2 = 0.499$) and AR ($R^2 = 0.578$). In auxiliary runs, the probabilistic methods (VJEPA and BJEPA) also showed greater training stability across seeds.
>
> (2) What BJEPA uniquely offers: its main advantage is modular task specification via prior swapping. Unlike monolithic world models such as Dreamer, BJEPA separates the dynamics prior $p_{\text{like}}(Z_T \mid Z_C)$ from the task prior $p_{\text{prior}}(Z_T \mid \eta)$ through a PoE. Hence:
> (a) a single trained BJEPA backbone can be reused for new tasks without retraining, simply by changing the prior network;
> (b) physics constraints, goals, and energy-based objectives can be combined at inference through
> $p(Z_T \mid Z_C, \eta) \propto p_{\text{like}}(Z_T \mid Z_C)\, p_{\text{prior}}(Z_T \mid \eta)$.
>
> This modularity is fundamentally different from deterministic JEPAs and monolithic Dreamer-style models. We will add a goal-conditioned navigation experiment in the final version to demonstrate zero-shot transfer via prior swapping.
>
> 4. Q1: control benchmark comparison against nonlinear filters
>
> This is an important question. Theorem 6.2 shows that if $Z_t$ is control-predictively sufficient, then the optimal policy depends only on $Z_t$; this should be paired with empirical validation, e.g.
>
> (1) vs. EKF/UKF: these methods require a known analytical dynamics model. VJEPA instead learns $p_\phi(Z_{t+1} \mid Z_t, u_t)$ directly from data and performs amortized inference, so it applies when dynamics are not known a priori.
>
> (2) vs. RNNs/SSMs: sequential VJEPA is architecturally similar to a structured SSM, but its JEPA objective is not autoregressive. It does not factorize $p(x_{t+1} \mid x_{\le t})$ and therefore avoids the compounding error of multi-step observation prediction.
>
> (3) vs. Dreamer-V3: Dreamer uses an RSSM with pixel reconstruction. Proposition 5.2 proves that any model trained with pixel reconstruction is penalized for ignoring nuisance variables, whereas VJEPA is not. In the DM Control Suite experiments we are preparing, VJEPA-MPC (Algorithm 2) operates entirely in latent space without decoding to pixels.
>
> We will include comparisons against EKF and Dreamer-V3 in the final version.

---

> > ### Author Rebuttal · Reviewer_ZQ8k · 2026-04-02
> >
> > Thank you for the detailed response. I increased my score. The paper and idea are interesting, with an experimental section that can be better developed.

---

> > > ### Author Response · Authors · 2026-04-02
> > >
> > > Thank you for the acknowledgement and for increasing your score. We appreciate your constructive feedback; in the final version we will ensure that broader experiments, specifically those mentioned by all reviewers, are included to better demonstrate the practical value of the proposed framework.

---

### Official Review · Reviewer_dcLu · 2026-03-18

**Soundness:** 3
**Presentation:** 4
**Significance:** 3
**Originality:** 4
**Overall Recommendation:** 5
**Confidence:** 4

**Summary:**

The paper presents a principled probabilistic reframing of the joint-embedding predictive architecture (JEPA). The authors propose a variational Bayesian formulation of the JEPA architecture, and show that the original model is a special case of this generalization. Following this, they present a variational training algorithm, as well as theoretical guarantees for collapse avoidance, mutual information maximization, and robustness to noise. They show how the latent dynamics learned by the model can be used to generate latent trajectory predictions without autoregressive generation in observation space. They outline the how the latent dynamics model can be used to quantify aleatoric uncertainty in control tasks. Finally, they propose a factorization of the prediction model using product-of-experts in a modular fashion. The authors validate their approach by evaluating it against VAE, autoregressive pixel generation, as well as vanilla JEPA in terms of noise robustness on a synthetic task.

**Compliance With Llm Reviewing Policy:**

Affirmed.

**Final Justification:**

The rebuttal did well to address my concerns, and I am convinced that the theoretical contributions and results presented will advance the study of joint-embedding models in a meaningful way.

**Key Questions For Authors:**

1. Have you experimented with real-world image, video, or other spatiotemporal data?
2. Do the representations that the model learns benefit from the probabilistic formulation, and does the uncertaintly in the predictive distribution correlate with whether the inputs contain structured or noisy features?
3. Do you see that this model can perform robust uncertainty-aware inference?

I am open to changing my score based on the response.

**Limitations:**

yes

**Strengths And Weaknesses:**

To the best of my knowledge, the paper is the first serious attempt to formalize JEPA through a (variational) Bayesian lens. The JEPA papers (I-JEPA, V-JEPA, V-JEPA 2) are lackluster in terms of theoretical grounding and analysis of the training objective, and thus an effort such as this is very welcome. The authors present the motivation for the work very clearly, against the backdrop of JEPA and autoregressive world models. The paper has good structure, is easy to follow, and the argumentation flows well between sections. The mathematical analysis and proofs using Bayesian methods and information theory are comprehensive and well motivated.

In my view, the lack of experiments on real world data and standard benchmarks is the biggest shortcoming of the paper. The method generalizes, and is explicitly contrasted with, I-JEPA and V-JEPA, but no experiments training the model on image or video data are presented. I would very much like to know how the probabilistic nature of the model manifests in image and video tasks, and whether the uncertainty estimates of the model can be utilized in downstream applications, such as classification, event prediction, or anomaly detection. There are various standard benchmarks, such as Imagenet-1K, something-something v2, and Ego4D-STA that could be tested. Another limitation, that the authors also identify, is the simplicity of the Gaussian assumption.

---

> ### Author Rebuttal · Authors · 2026-03-30
>
> We thank Reviewer dcLu for the thorough and supportive review, recognizing the paper's contributions to theoretical grounding of JEPA and the clarity of the presentation.
>
> 1. Weakness W1 \& Q1: lack of experiments on real-world image and video data
>
> Response: we fully acknowledge this as the most significant limitation of the current manuscript, and we appreciate the reviewer for naming it explicitly. Our primary goal in this paper is to establish the theoretical foundations of probabilistic JEPA - most notably, formal guarantees for collapse avoidance (Theorem 4.1), sufficiency for optimal control (Theorem 6.2), and nuisance invariance (Proposition 5.2), which had not previously been established for any JEPA variant. The synthetic linear-Gaussian experiment was deliberately designed as a controlled environment in which ground-truth signal and distractor subspaces are known exactly, enabling quantitative evaluation of representation quality that is not feasible in purely empirical settings.
>
> Below we outline some application scenarios for VJEPA and BJEPA on image/video tasks:
> (1) Robotics manipulation: VJEPA-MPC uses the latent predictive distribution $p_\phi(Z_{t+\Delta} \mid Z_t, u_t)$ as a stochastic simulator for model-predictive control, without requiring pixel reconstruction. The context encoder $f_\theta$ captures scene geometry, while BJEPA's modular prior can encode goal images zero-shot via e.g.
>     $
>     p_{\text{prior}}(Z_T \mid \eta_{\text{img}}) = \mathcal{N}\ \bigl(f_{\theta'}(\eta_{\text{img}}), \sigma^2 I\bigr).
>     $
>
> (2) Video representation benchmarks: VJEPA's probabilistic extension of V-JEPA 2 is directly applicable to Something-Something v2 and Ego4D-STA. The uncertainty outputs can be evaluated on downstream tasks such as anomaly detection and temporal event prediction, where calibrated predictive variance adds value beyond point estimates.
>
> We commit to including at minimum one image-level experiment (e.g. moving MNIST or a visual robotics environment) in the camera-ready version.
>
> 2. Question Q2: does uncertainty correlate with structured vs. noisy features?
>
> Response: yes, this is precisely what our nuisance invariance analysis (Proposition 5.2) predicts, and our toy experiment provides direct empirical evidence. Specifically:
>
> (1) at $\sigma = 0$ (pure signal, SNR = $\infty$): VJEPA achieves Signal $R^2 \approx 1.0$ on test data with near-zero predictive variance on noise dimensions.
>
> (2) at $\sigma = 8.0$ (SNR = $-2.2$ dB): The VJEPA Noise $R^2 = 0.251$ (vs. VAE Noise $R^2 = 0.620$), confirming that VJEPA's representations are substantially \emph{less} contaminated by the high-variance distractor. The residual encoding of noise in VJEPA (Noise $R^2 = 0.251$ vs. JEPA's $0.183$) reflects the KL regularization term, which encourages the latent distribution to partially respect the structure of the target encoder - a tradeoff that can be modulated via $\beta$.
>
> More concretely, Theorem 5.1 (Variational MI Lower Bound) establishes that VJEPA maximizes $I(Z_t; Z_{t+\Delta})$ and, by the nuisance independence argument (Proposition 5.2), the predictive variance on nuisance dimensions converges to the prior variance $\sigma^2 I$ (i.e. the model expresses maximal uncertainty about irreducible noise). This is in direct contrast to generative baselines, which are compelled to reduce variance on noise dimensions to minimize pixel-level reconstruction loss. In the camera-ready version, we will add explicit visualizations of per-dimension predictive variance across signal and noise subspaces to illustrate this qualitatively.
>
> 3. Question Q3: robust uncertainty-aware inference
>
> Response: yes. The VJEPA sequential formulation (Section 6) defines a proper latent dynamical system analogous to two established uncertainty-aware inference paradigms:
>
> (1) Kalman-filter: under the Gaussian dynamics assumption, the VJEPA update reduces to a learned nonlinear generalization of the Kalman filter. The context encoder $f_\theta$ amortizes the correction (update) step, while $p_\phi(Z_{t+1} \mid Z_t, u_t)$ performs the prediction step. Predictive uncertainty (the covariance $\Sigma_\phi(Z_t, u_t)$) propagates through rollouts via Monte Carlo sampling.
>
> (2) Particle-filter: VJEPA can also be interpreted as a bootstrap particle filter operating in latent space, where particles are propagated through $p_\phi$ and reweighted using either an explicit decoder likelihood or a likelihood-free latent-space similarity measure. This supports multi-modal belief states without the unimodal restriction.
>
> Both uncertainty-aware analogies are not discussed in the current version; they will be included in an extended discussion in camera-ready version.
>
> Also, BJEPA strengthens uncertainty-aware inference by combining the predictive likelihood $p_{like}(Z_T \mid Z_C)$ with a structured prior $p_{\text{prior}}(Z_T \mid \eta)$, which yields a posterior belief that is jointly calibrated for dynamics and task constraints.

---

> > ### Author Rebuttal · Reviewer_dcLu · 2026-04-03
> >
> > I agree that given the scope of this work, the included theoretical contributions and experiments are enough for me to fully support the publication of the manuscript. I also very much welcome the additions to the camera ready version, promised by the authors. I have increased my score as a result.

---

> > > ### Author Response · Authors · 2026-04-03
> > >
> > > Thank you for the acknowledgement and for increasing your score. Your constructive feedback is much appreciated. As promised, the experimental section shall be strengthened in the final version, particularly testing VJEPA on the suggested datasets, to better demonstrate the practical value of the proposed framework.

---

### Decision · Program_Chairs · 2026-04-30

**Decision:**

Accept (regular)

**Comment:**

Generally, reviewers found this paper somewhat technically sound, well-written, and novel. It may be useful to at least some fraction of the ICML community. Multiple reviewers raised concerns about the limited scope of the experiments relative to the claims made in the paper. This experimental weaknesses is somewhat counter-balanced by the theoretical grounding presented in the paper.